# Metric Learning from Limited Pairwise Preference Comparisons

**Zhi Wang**[†1]      **Geelon So**[2]      **Ramya Korlakai Vinayak**[1]

[1] Dept. of Electrical and Computer Engineering, University of Wisconsin-Madison
[2] Dept. of Computer Science and Engineering, University of California San Diego
zhi.wang@wisc.edu, geelon@ucsd.edu, ramya@ece.wisc.edu

## Abstract

We study metric learning from preference comparisons under the ideal point model, in which a user prefers an item over another if it is closer to their latent ideal item. These items are embedded into $\mathbb{R}^d$ equipped with an unknown Mahalanobis distance shared across users. While recent work shows that it is possible to simultaneously recover the metric and ideal items given $\mathcal{O}(d)$ pairwise comparisons per user, in practice we often have a limited budget of $o(d)$ comparisons. We study whether the metric can still be recovered, even though it is known that learning individual ideal items is now no longer possible. We show that in general, $o(d)$ comparisons reveal no information about the metric, even with infinitely many users. However, when comparisons are made over items that exhibit low-dimensional structure, each user can contribute to learning the metric restricted to a low-dimensional subspace so that the metric can be jointly identified. We present a divide-and-conquer approach that achieves this, and provide theoretical recovery guarantees and empirical validation.

## 1 INTRODUCTION

Metric learning is commonly used to discover measures of similarity for downstream applications [e.g., Kulis et al., 2013]. In this paper, we study metric learning from pairwise preference comparisons. In particular, we consider the ideal point model [Coombs, 1950], in which a set of items are embedded into $\mathbb{R}^d$, and a user prefers an item $x$ over another $x'$ if it is *closer* to the user's latent ideal point $u \in \mathbb{R}^d$, i.e.,

$$\rho(x, u) < \rho(x', u),$$

for some underlying metric $\rho : \mathbb{R}^d \times \mathbb{R}^d \to \mathbb{R}_{\geq 0}$. While

---

†Work done at University of California San Diego.

high-quality item embeddings have become increasingly available, for example from foundation models pre-trained on internet-scale data [e.g., Radford et al., 2021], naively equipping these representations with the Euclidean distance may not accurately capture the semantic relations between items as perceived by humans, and therefore may not align with human values or preferences [Yu et al., 2014, Canal et al., 2022]. Meanwhile, people often agree on their perception of item similarities [Colucci et al., 2016]. In this work, we study when and how a shared Mahalanobis distance can be learned from a large crowd, where each user answers a few queries of the form: "Do you prefer $x$ or $x'$?"

The line of work on simultaneous metric and preference learning was recently introduced by Xu and Davenport [2020], who studied it under the ideal point model for a single user. They proposed an alternating minimization algorithm to recover both the Mahalanobis distance and user ideal point. After, Canal et al. [2022] introduced a convex formulation of the problem, providing the first theoretical guarantees while extending the results to crowdsourced data. They showed that the cost of learning a Mahalanobis distance can be amortized among users; it is possible to jointly learn the metric and ideal points in $\mathbb{R}^d$ so long as sufficiently many users each provides $\Theta(d)$ preference comparisons.

However, when the representations of data are very high-dimensional, obtaining $\Omega(d)$ preference comparisons from each user can be practically infeasible. It can be expensive to ask a user more than a few queries [Cohen et al., 2005] both in terms of cost and cognitive overload, and users may have concerns over their privacy [Jeckmans et al., 2013]. Fortunately, through crowdsourcing, we often have access to preference comparisons from a *large* pool of users. In this paper, we ask the fundamental question:

> *Can we learn an unknown Mahalanobis distance in $\mathbb{R}^d$ from $o(d)$ preference comparisons per user?*

We provide a twofold answer to this question. First, we show a negative result: even with infinitely many users, it is gener-

ally impossible to learn anything at all about the underlying metric when each user provides fewer than $d$ preference comparisons. In general, there is no hope for recovering the unknown metric from preference comparisons without learning individual preference points as well.

Second, we show that the negative result does not rule out the possibility of learning the metric when the set of items are *subspace-clusterable* (Definition 12); that is, when they lie in a union of low-dimensional subspaces [Parsons et al., 2004, Ma et al., 2008]. These subspaces may capture, for instance, different categories or classes of items [Elhamifar and Vidal, 2013]. Such structure has also been studied extensively in compressed sensing [Lu and Do, 2008, Eldar and Mishali, 2009] and face recognition [Basri and Jacobs, 2003, Ho et al., 2003], among others. Given items with subspace-clusterable structure, we show that we can learn the Mahalanobis distance using a *divide-and-conquer* approach (Figure 1). This involves learning the metric restricted to each subspace, which is feasible using very few comparisons per user, and then reconstructing the full metric from these subspace metrics.

**Contributions**  We study the fundamental problem of learning an unknown metric with limited pairwise comparison queries, i.e, whether it is possible to learn a shared unknown metric without learning the individual preference points. Our main contributions are as follows:

1. We provide an impossibility result: nothing can be learned if the items are in general position (Section 3);

2. We define the notion of subspace-clusterable items and propose a divide-and-conquer approach, such that:

   - Given noiseless, unquantized comparisons that indicate how much a user prefers one item over another, we show that subspace-clusterability is necessary and sufficient for identifying the unknown metric (Section 4);

   - Given noisy, quantized comparisons in the form of binary responses over subspace-clusterable items, we present recovery guarantees in terms of identification errors for our approach (Section 5);

3. We implement our proposed algorithm and validate our findings using synthetic data (Section 6).

## 1.1  RELATED WORK

There is a rich literature on metric learning; see [Kulis et al., 2013] for a survey. A line of metric learning from human feedback focuses on learning Mahalanobis distances from triplet comparisons [Schultz and Joachims, 2003, Verma and Branson, 2015, Mason et al., 2017], in which users are asked "is $u$ closer to $x$ or $x'$?" However, triplet comparisons are a specific type of feedback that is not always practical to obtain. And so, an important extension of these works is

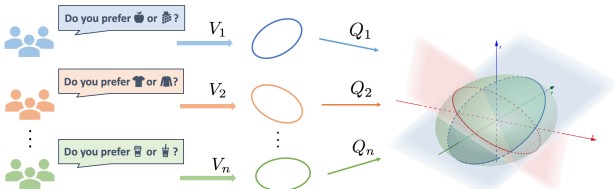

Figure 1: In our divide-and-conquer approach, users help us recover the metric $Q_\lambda$ restricted to subspaces $V_\lambda$. We stitch these together to recover the metric $M$ on $\mathbb{R}^d$. The ellipses visualize the low-dimensional unit spheres, which are "slices" of the full metric.

metric learning from preference comparisons, which can be seen as a variant of triplet comparisons with an unknown latent comparator $u$. Even though preference comparisons are a weaker form of feedback, they are also much more prevalent. For example, they can be inferred from user behavior, assuming users tend to engage more with items perceived to be more ideal.

Specifically, in this paper, we consider the ideal point model [Coombs, 1950], where the latent comparator $u$ in a preference comparison represents a user's ideal item. Note that, if the ideal points are known beforehand, one can simply treat this as a problem of metric learning from triplet comparisons. Conversely, if the metric is known, one can also localize user ideal points using techniques from [Jamieson and Nowak, 2011, Massimino and Davenport, 2021, Tatli et al., 2024].

Our paper builds upon recent research that studies *simultaneous* metric and preference learning [Xu and Davenport, 2020, Canal et al., 2022]. In a single user setting, Xu and Davenport [2020] developed an algorithm that iteratively alternate between estimating the metric and the user ideal point. Canal et al. [2022] generalized the setting to involve multiple users. They established identifiability guarantees when users provide unquantized measurements, and presented generalization bounds and recovery guarantees when users provide binary responses. While Canal et al. [2022] showed that it is possible to jointly recover a metric and user ideal points when each user answers $\Theta(d)$ queries, we address the fundamental question of learning Mahalanobis distances when we have a much limited budget of $o(d)$ preference comparisons per user. The $o(d)$ budget is more realistic especially when items are embedded in higher dimensions, but also poses interesting new challenges as learning user ideal points is no longer possible.

Several other works in the broader literature are related. For example, learning ordinal embeddings or kernel functions from triplet comparisons has been studied. Tamuz et al. [2011] developed an active multi-dimensional scaling algorithm to learn item embeddings, with the goal of capturing item similarities perceived by humans. See also [Van

Der Maaten and Weinberger, 2012, Jain et al., 2016, Klein-dessner and von Luxburg, 2017], among other works. Hsieh et al. [2017] introduced a collaborative metric learning algorithm, which uses matrix factorization to learn user and item embeddings such that the Euclidean distance reflects user preferences and item/user similarities. A divide-and-conquer approach for deep metric learning has been studied by Sanakoyeu et al. [2019], who use $k$-means to cluster items and learn separate metrics for each cluster before concatenating them together; they performed an extensive empirical study based on image data. In comparison, we consider the ideal point model to study the fundamental problem of metric learning from limited preference comparisons. As we build directly on this line of work by Xu and Davenport [2020] and Canal et al. [2022], we now present background and existing results in greater detail.

## 2 PRELIMINARIES

**The ideal point model** Let $\mathcal{X}$ be a set of items embedded into $\mathbb{R}^d$ with an unknown Mahalanobis distance $\rho$. Let $M$ be its matrix representation in $\mathbb{R}^{d \times d}$. That is, $M$ is a positive-definite (symmetric) matrix and for all $x, x' \in \mathbb{R}^d$,

$$\rho(x, x') := \sqrt{(x - x')^\top M (x - x')} = \|x - x'\|_M.$$

Suppose there is a large pool of users, and each user is associated with an unknown ideal point in $\mathbb{R}^d$. A user with ideal point $u$ prefers an item $x$ over another $x'$ if and only if $\rho(x, u) < \rho(x', u)$; or whenever $\psi(x, x'; u) < 0$, where:

$$\psi_M(x, x'; u) := \|x - u\|_M^2 - \|x' - u\|_M^2. \quad (1)$$

Each user's ideal point may be distinct, but we assume that the metric $\rho$ is a shared. We aim to recover $\rho$ when each user provides very few preference comparisons.

We consider two types of user preference comparisons for learning the metric: *unquantized* and *quantized measurements*. From a user with ideal point $u$, these are of the form:

$$\underbrace{(x, x', \psi)}_{\text{unquantized}} \quad \text{and} \quad \underbrace{(x, x', y)}_{\text{quantized}},$$

where $\psi = \psi(x, x'; u)$ is a real number that indicates the difference between the squared distances, and $y$ is binary, taking values in $\{-1, +1\}$. When $y = -1$, $x$ is preferred over $x'$, and $y = +1$ indicates otherwise.

**Metric learning from preference measurements** We now review the existing algorithmic ideas for recovering the metric from preference feedback under the ideal point model. Suppose that we are given unquantized measurements from a single user with an ideal point $u \in \mathbb{R}^d$. With a little algebra [Canal et al., 2022], the measurement in Eq. (1) becomes:

$$\psi_M(x, x'; u) = \langle xx^\top - x'x'^\top, M \rangle + \langle x - x', v \rangle, \quad (2)$$

where $v := -2Mu$. The first inner product is the trace inner product for matrices, while the second inner product is the usual inner product on $\mathbb{R}^d$. The re-parametrization $v$ of $u$ is sometimes called the *pseudo-ideal point*. Thus, unquantized measurements are linear over the joint variables $(M, v)$. Given a set of unquantized measurements from a user, one can just solve a linear system of equations to recover the matrix representation $M$ of $\rho$, as described in Algorithm 3 of Appendix A. Since $M$ has full rank and therefore invertible, we can then recover $u$ from $M$ and $v$ [Canal et al., 2022].

As there are $\frac{d(d+1)}{2} + d$ degrees of freedom in $(M, v)$, to recover the metric in this way requires at least that many measurements from a single user; the first term corresponds to the dimension of symmetric $d \times d$ matrices representing Mahalanobis distances, the second for the user ideal point.

When $d^2$ is very large, we may want to amortize learning the metric over many users. Canal et al. [2022] show that this is possible. Let the users be indexed by elements in $[K]$. We can construct a larger linear regression problem, where each user has a separate covariate corresponding to their ideal point. Now, the joint variable is $(M, v_1, v_2, \ldots, v_K)$, which has $\frac{d(d+1)}{2} + dK$ degrees of freedom. When the population is large, it suffices to ask each user $\Theta(d + d^2/K)$ preference queries, which can be much closer to $d$ than $d^2$. This procedure is given in Algorithm 4 of Appendix A.

However, modern representations of data may be extremely high-dimensional, and it would be too onerous for any single user to provide $d$ measurements. In this paper, we tackle this question: If we have access to many users but can only ask each user a much more limited number $m \ll d$ of preferences queries, can we still recover $\rho$? We note that with $o(d)$ pairwise queries, it is impossible to localize the ideal preference point of a user even with a known metric [Jamieson and Nowak, 2011, Massimino and Davenport, 2021]. So, our goal here is to address the open question of whether it is possible to learn an *unknown* metric with such limited queries per users given a sufficiently large pool of users.

**Notation** Let $\mathrm{Sym}(\mathbb{R}^d)$ denote the symmetric $d \times d$ matrices equipped with the trace inner product, and let $\mathrm{Sym}^+(\mathbb{R}^d)$ be the positive-definite matrices. For readability, we often make abbreviations of the form $\Delta \in \mathrm{Sym}(\mathbb{R}^d)$ and $\delta \in \mathbb{R}^d$:

$$\Delta \equiv xx^\top - x'x'^\top \quad \text{and} \quad \delta \equiv x - x'.$$

Then, $\Delta \oplus \delta$ is an element of $\mathrm{Sym}(\mathbb{R}^d) \oplus \mathbb{R}^d$, the direct sum of inner product spaces, and we can shorten Eq. (2) to:

$$\psi_M(x, x'; u) = \langle \Delta \oplus \delta, M \oplus v \rangle.$$

Following the experimental design literature, let us call a collection of such elements a *design matrix*:

**Definition 1.** *Let* $\{(x_{i_0}, x_{i_1})\}_{i \in [m]}$ *be a collection of item pairs. It induces the linear map* $D : \mathrm{Sym}(\mathbb{R}^d) \times \mathbb{R}^d \to \mathbb{R}^m$,

$$D(A, w)_i = \langle \Delta_i \oplus \delta_i, A \oplus w \rangle,$$

where $\Delta_i = x_{i_0} x_{i_0}^\top - x_{i_1} x_{i_1}^\top$ and $\delta_i = x_{i_0} - x_{i_1}$ for $i \in [m]$. *As a slight abuse of language, we call $D$ the induced* design matrix. *If item pairs are drawn from a distribution $\mathcal{P}_m$ over $(\mathbb{R}^d \times \mathbb{R}^d)^m$, we say that $D$ is a* random design *and write $D \sim \mathcal{P}_m$. We also define $\sigma_{\min}^2(\mathcal{P}_m) = \frac{1}{m} \cdot \sigma_{\min}(\mathbb{E}[D^*D])$.*

For additional background and notation, see Appendix B.

## 3   AN IMPOSSIBILITY RESULT

Consider the mathematically simplified setting in which users provide *unquantized* responses. We show a negative result stating that when users provide fewer than $d$ comparisons, we fundamentally cannot learn anything about $M$ if the items are in general position in the following sense:

**Definition 2.** *A set $\mathcal{X} \subset \mathbb{R}^d$ has* generic pairwise relations *if for any acyclic graph $G = (\mathcal{X}, E)$ with at most $d$ edges, the set $\{x - x' : (x, x') \in E\}$ is linearly independent.*

The geometric meaning of having generic pairwise relations is simple: if any $d$ pairs of points are connected by lines, then those lines are linearly independent (unless they form cycles; see Figure 2 for an illustration). Proposition C.3 shows that almost all finite subsets of Euclidean space have generic pairwise relations with respect to the Lebesgue measure[1].

The following theorem shows that if items have generic pairwise relations, then sets of $m \leq d$ unquantized measurements from a single user provide no information about the underlying metric. In particular, suppose that $M$ and $v$ are the underlying matrix representation and user's pseudo-ideal point, both unknown to us. Then, for any other Mahalanobis matrix $M'$, we can find a pseudo-ideal point $v'$ that is also consistent with the data. In fact, the negative result holds even with infintely many users:

**Theorem 3.** *Fix $M \in \mathrm{Sym}^+(\mathbb{R}^d)$ and $v_k \in \mathbb{R}^d$ for each $k \in \mathbb{N}$. Let $(D_k)_{k \in \mathbb{N}}$ be a collection of design matrices, each for a set of $m \leq d$ pairwise comparisons. If each set of compared items has generic pairwise relations, then for all $M' \in \mathrm{Sym}^+(\mathbb{R}^d)$, there exists $(v'_k)_{k \in \mathbb{N}} \subset \mathbb{R}^d$ such that:*

$$D_k(M, v_k) = D_k(M', v'_k), \qquad \forall k \in \mathbb{N}.$$

See Appendix C for a proof of Theorem 3. This theorem shows that when items have *generic pairwise relations* it is not just that we cannot recover $\rho$, but that we cannot glean anything at all about $\rho$ when users each provide $d$ or fewer comparisons, for every matrix in $\mathrm{Sym}^+(\mathbb{R}^d)$ is consistent with $D$. While each user provides us with more data, each also introduces new degrees of freedom—the unknown ideal points. When learning from crowds, more data does not necessarily lead to more usable information.

---

[1]In Appendix C, we discuss the connection between generic pairwise relations and general linear position, a standard notion from geometry.

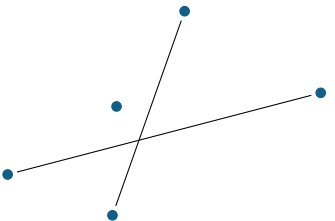

Figure 2: Points in $\mathbb{R}^2$ with generic pairwise relations.

## 4   EXACT RECOVERY WITH LOW-RANK SUBSPACE STRUCTURE

The above negative result applies to almost all finite sets of items. It seems to tell a pessimistic story for metric learning when data is embedded into high dimensions and when it is infeasible to obtain $\Omega(d)$ preference comparisons per user.

However, the story is not closed and shut yet. Real-world data often exhibit additional structure that could help us recover the metric, such as low intrinsic dimension [Fefferman et al., 2016]. In particular, we assume that many items of $\mathcal{X}$ lie on a *union of subspaces*. The approximate validity of this assumption is the basis of work in manifold learning [Roweis and Saul, 2000, Tenenbaum et al., 2000, Belkin and Niyogi, 2003], compressed sensing [Donoho, 2006], and sparse coding [Olshausen and Field, 1997], among others.

In this case, we can take a divide-and-conquer approach to metric learning by identifying the metric restricted to those subspaces, before stitching them back together to recover the full metric. Let's define subspace Mahalanobis distances:

**Definition 4.** *Let $V$ be a subspace of $\mathbb{R}^d$. A metric on $V$ is a* subspace Mahalanobis distance *if it is a subspace metric of some Mahalanobis distance $\rho$ on $\mathbb{R}^d$. In that case, we denote the subspace metric by $\rho\big|_V$, where for all $x, x' \in V$,*

$$\rho\big|_V(x, x') = \rho(x, x').$$

In general, we cannot hope to identify an arbitrary metric from a finite number of its subspace metrics. However, Mahalanobis distances have much more structure than arbitrary metrics on $\mathbb{R}^d$. A Mahalanobis distance on $\mathbb{R}^d$ can be fully specified using $d(d+1)/2$ numbers. By recovering its subspace metrics, we can hope to chip away at the degrees of freedom of Mahalanobis distances.

As another way of intuition, each Mahalanobis distance may be identified with its unit sphere—points that are unit distance away from the origin. These points form a $(d-1)$-dimensional ellipsoid in $\mathbb{R}^d$. To recover a subspace Mahalanobis distance on $V$ means that we are able to determine which points of $V$ intersect this ellipsoid (see Figure 1). If we do this for sufficiently many subspaces, we can determine the whole ellipsoid. To formalize this intuition, we

now linear-algebraically relate a Mahalanobis distance with its subspace metrics.

## 4.1 A LINEAR PARAMETRIZATION OF MAHALANOBIS DISTANCES

To describe the linear relationship between a Mahalanobis distance and its subspace metrics, we need to parametrize the subspace metrics. To do so, we first need to fix a choice of coordinates on each $V \subset \mathbb{R}^d$. In the following, let $V$ be an $r$-dimensional subspace of $\mathbb{R}^d$ and let $B \in \mathbb{R}^{d \times r}$ be an orthonormal basis of $V$, where $r \ll d$.

**Definition 5.** *We say $V$ has a* canonical representation *if it is equipped with an orthonormal basis $B$, where the canonical representation of a vector $x \in V$ is given by $B^\top x \in \mathbb{R}^r$.*[2]

**Definition 6.** *Let $\mathrm{Sym}(V)$ and $\mathrm{Sym}^+(V)$ respectively denote the pairs $(\mathrm{Sym}(\mathbb{R}^r), B)$ and $(\mathrm{Sym}^+(\mathbb{R}^r), B)$, where $V$ has a canonical representation given by $B$.*

*Let $Q \in \mathrm{Sym}(V)$ mean that $Q \in \mathrm{Sym}(\mathbb{R}^r)$, and that it carries the basis information $B$ along with it.*

Just as Mahalanobis distances on $\mathbb{R}^d$ are in one-to-one correspondence with positive-definite matrices, so too are Mahalanobis distances on $V$ in correspondence with $\mathrm{Sym}^+(V)$. Furthermore, Proposition D.1 shows that the matrix representations of a Mahalanobis distance and its restriction to a subspace is given by the following linear map.

**Definition 7.** *Let $V$ and $B$ be as before. Define the linear map $\Pi_V : \mathrm{Sym}(\mathbb{R}^d) \to \mathrm{Sym}(V)$ by:*

$$\Pi_V(A) = B^\top A B. \tag{3}$$

Thus, if a Mahalanobis distance $\rho$ on $\mathbb{R}^d$ and its restriction $\rho\big|_V$ to a subspace $V$ have representations $M \in \mathrm{Sym}^+(\mathbb{R}^d)$ and $Q \in \mathrm{Sym}^+(V)$, respectively, then:

$$Q = \Pi_V(M) = B^\top M B.$$

## 4.2 LEARNING WITH LOW-RANK SUBSPACES

To see how low-dimensional structure can help us make progress in learning the metric, consider a simple setting where all items lie in some low-dimensional subspace $V$. Instead of learning the full metric $\rho$, we could aim for a more modest goal of learning the subspace metric $\rho\big|_V$.

As before, let $V$ be an $r$-dimensional subspace of $\mathbb{R}^d$ with a canonical representation. If all items and ideal points lie in $V$, then learning $\rho\big|_V$ immediately reduces to the usual setting of learning a Mahalanobis distance, since we can

---

[2] We shall always equip $\mathbb{R}^d$ with the standard basis, so that a vector is its own canonical representation in $\mathbb{R}^d$.

simply ignore the remaining dimensions and reparametrize the problem. But when the ideal points are not assumed to lie on $V$, it is not evident *a priori* that we can ignore the dimensions extending beyond the set of items. However, it turns out that for Mahalanobis distances, we may.

The next lemma shows that even if a user's ideal point $u \in \mathbb{R}^d$ falls outside of $V$, for items in $V$, there is a phantom ideal point $u_V \in V$ such that preference comparisons for items in $V$ generated by $u$ and $u_V$ are equivalent.

**Lemma 8.** *Let $V$ be an $r$-dimensional subspace of $\mathbb{R}^d$ with a canonical representation given by $B \in \mathbb{R}^{d \times r}$. Fix any Mahalanobis distance $M \in \mathrm{Sym}^+(\mathbb{R}^d)$, any pair of items $x, x' \in \mathbb{R}^d$, and ideal point $u \in \mathbb{R}^d$. Suppose that $x$ and $x'$ are contained in $V$ with canonical representation $x_V = B^\top x$ and $x_V' = B^\top x'$ in $\mathbb{R}^r$. Then:*

$$\psi_M(x, x'; u) = \psi_Q(x_V, x_V'; u_V),$$

*where the phantom ideal point $u_V$ of $u$ on $V$ satisfies $(B^\top M B) u_V = B^\top M u$, and $Q = \Pi_V(M)$ is the matrix representation in $\mathrm{Sym}^+(V)$ of the subspace metric $\rho\big|_V$.*

Consequently, learning a subspace metric $\rho\big|_V$ turns into a problem of metric learning from preference comparisons in $\mathbb{R}^r$. From here, we can simply use existing algorithms to recover the matrix representation of the subspace metric. By Canal et al. [2022], it is possible to identify the subspace metric so long as users can each provide $m \geq \Omega(r)$ preference comparisons. For this easier problem of learning $\rho\big|_V$, when $r \ll d$, we can do with $o(d)$ responses per user.

In the remainder of this section, we give a simple characterization for when a Mahalanobis distance on $V$ can be learned from preference comparisons of items on $V$. The set of items needs to be sufficiently rich so that all degrees of freedom of $\mathrm{Sym}(V) \oplus V$ can be captured. We define:

**Definition 9.** *Let $V$ be a subspace of $\mathbb{R}^d$ with canonical representation given by $B$. A subset $\mathcal{X}_V \subset V$ quadratically spans $V$ if $\mathrm{Sym}(V) \oplus V$ is linearly spanned by the set:*

$$\left\{ (x_V x_V^\top - x_V' x_V'^\top) \oplus (x - x') : x, x' \in \mathcal{X}_V \right\},$$

*where $x_V = B^\top x$ and $x_V' = B^\top x'$ denote the canonical representations of $x$ and $x'$ in $V$.*

If we have no restriction on how many queries we can ask a user, then it is straightforward to see that quadratic spanning is a sufficient condition for recovering the underlying metric. For simplicity, let $V = \mathbb{R}^d$. If $\mathcal{X}$ quadratically spans $\mathbb{R}^d$, then we can detect all dimensions of $M \oplus v$ corresponding to the Mahalanobis matrix and a user's pseudo-ideal point. To do so, choose any design matrix $D : \mathrm{Sym}(\mathbb{R}^d) \oplus \mathbb{R}^d \to \mathbb{R}^m$ whose rows $\{\Delta_i \oplus \delta_i : i \in [m]\}$ span $\mathrm{Sym}(\mathbb{R}^d) \oplus \mathbb{R}^d$.

When the number of queries is limited per user, the following result shows that the quadratic spanning condition is

**Algorithm 1:** Metric learning from subspace clusters

**Input:** Unquantized measurements over items that lie in a union of subspaces $V_\lambda, \lambda \in \Lambda$

// Stage 1: learning subspace metrics

1 **for** *each subspace* $\lambda \in \Lambda$ **do**

2  Recover $\hat{Q}_\lambda \in \mathrm{Sym}(\mathbb{R}^{r_\lambda})$ with respect to $B_\lambda$ via reduction to Algorithm 4 [Canal et al., 2022]

// Stage 2: reconstruction

3 Solve the linear equations over $A \in \mathrm{Sym}(\mathbb{R}^d)$:

$$B_\lambda^\top A B_\lambda = \hat{Q}_\lambda, \quad \lambda \in \Lambda$$

**Output:** $\hat{A}$, the solution to the above linear equations.

---

still sufficient for recovering $\rho\big|_V$, provided we can ask many users $m \geq \dim(V) + 1$ unquantized preference queries.

**Proposition 10.** *Let $\mathcal{X}$ quadratically span a subspace $V$ of dimension $r$. There exists a collection $D_1, \ldots, D_K$ of design matrices, each over $m$ pairs of items in $\mathcal{X}$, such that given a (distinct) user's response to each design, $\rho\big|_V$ can be identified when $m \geq r + 1$ and $K \geq r(r+1)/2$.*

To complement this sufficient condition, the next result shows that if $\mathcal{X}$ does not quadratically span $V$, then the subspace metric $\rho\big|_V$ cannot be recovered from only preference comparisons of items in $\mathcal{X} \cap V$.

**Proposition 11.** *Let $(D_k)_{k \in \mathbb{N}}$ be a set of design matrices over items in $\mathcal{X} \subset V$. If $\mathcal{X}$ does not quadratically span $V$, then infinitely many Mahalanobis distances on $V$ are consistent with any set of user responses to the design matrices.*

Proofs for the above results are deferred to Appendix D.2.

### 4.3 LEARNING WITH SUBSPACE-CLUSTERS

We have seen how to partially learn a Mahalanobis distance given many items within a subspace. We now consider how to fully recover the metric when many items lie in a *union of subspaces* $(V_\lambda)_{\lambda \in \Lambda}$. In this case, a divide-and-conquer approach is intuitive: (i) recover each subspace metric, then (ii) reconstruct $\rho$ from the learned subspace metrics. Recall that each subspace metric $\rho\big|_V$ is related to the full metric $\rho$ by the linear map $\Pi_V$ from Definition 7. Therefore, we can reconstruct $\rho$ from its subspace metrics by solving a system of linear equations. Algorithm 1 summarizes this approach.

In order to characterize when a Mahalanobis distance can be reconstructed from its subspace metrics, we introduce the notion of subspace-clusterability. A set of items $\mathcal{X}$ is subspace-clusterable when many of its items lie on sufficiently many item-rich subspaces. Formally:

**Definition 12.** *A set $\mathcal{X} \subset \mathbb{R}^d$ is subspace-clusterable over subspaces $V_\lambda \subset \mathbb{R}^d$ indexed by $\lambda \in \Lambda$ whenever:*

1. *each subset $\mathcal{X} \cap V_\lambda$ quadratically spans $V_\lambda$.*
2. *$\{xx^\top : x \in V_\lambda, \lambda \in \Lambda\}$ linearly spans $\mathrm{Sym}(\mathbb{R}^d)$.*

By Propositions 10 and 11, the first condition is necessary and sufficient for recovering each subspace metric $\rho\big|_{V_\lambda}$. Proposition 13 shows that the second condition is necessary and sufficient for recovering the $\rho$ from subspace metrics.

**Proposition 13.** *Let $\rho$ be a Mahalanobis distance on $\mathbb{R}^d$. Let $(V_\lambda)_{\lambda \in \Lambda}$ be a collection of subspaces with canonical representations given by the orthonormal bases $(B_\lambda)_{\lambda \in \Lambda}$. The following are equivalent:*

1. *$\{xx^\top : x \in V_\lambda, \lambda \in \Lambda\}$ spans $\mathrm{Sym}(\mathbb{R}^d)$.*
2. *Let $\Pi_{V_\lambda}$ be given by Equation (3). The linear map $\Pi : \mathrm{Sym}(\mathbb{R}^d) \to \bigoplus_{\lambda \in \Lambda} \mathrm{Sym}(V_\lambda)$ is injective, where:*

$$\Pi(A) = \bigoplus_{\lambda \in \Lambda} \Pi_{V_\lambda}(A).$$

3. *If $\hat{\rho}$ is a Mahalanobis distance such that $\hat{\rho}\big|_{V_\lambda} = \rho\big|_{V_\lambda}$ for all $\lambda \in \Lambda$, then $\hat{\rho} = \rho$.*

See Appendix D.3 for the proof. This proposition verifies the correctness of Algorithm 1. Let $Q_\lambda \in \mathrm{Sym}^+(V)$ represent $\rho\big|_{V_\lambda}$. Then, step 3 of the algorithm specifies that $\Pi_{V_\lambda}(A) = Q_\lambda$. If $\Pi$ is injective, then the only matrix $A \in \mathrm{Sym}(\mathbb{R}^d)$ consistent with the system of linear equations is the one that represents $\rho$.

**Remark 14.** *We can compute the number of subspaces required to identify $\rho$ using Proposition 13. For example, when $\dim(V_\lambda) = 1$ for each $\lambda \in \Lambda$, each subspace captures one degree of freedom of $\rho$, so $|\Lambda| \geq \frac{d(d+1)}{2}$ is necessary. See Figure D.1 in Appendix D for geometric intuition.*

## 5 APPROXIMATE RECOVERY FROM BINARY RESPONSES

Previously, we studied metric learning from unquantized preference comparisons of the form $(x, x', \psi)$. We now consider a more realistic setting where we obtain binary responses of the form $(x, x', y)$, where $y \in \{-1, +1\}$. Furthermore, we assume that responses are quantized and noisy, where noise can depend on the user and items, as in [Mason et al., 2017, Xu and Davenport, 2020, Canal et al., 2022].

For our divide-and-conquer approach, due to the inexactness of the responses, we can no longer expect to exactly identify each subspace metric. However, we show that as long as each subspace metric can be recovered approximately, then they can be stitched together to approximately recover the full metric (Theorem 15). And indeed, approximate recovery in each subspace is known to be possible. In Proposition 19, we present a version of Theorem 4.1 of Canal et al. [2022] adapted to subspaces; this gaurantee is provided under a probabilistic noise model that we describe shortly.

**Algorithm 2:** Metric learning from binary responses

**Input:** Quantized measurements over items that lie in a union of subspaces $V_\lambda, \lambda \in \Lambda$

// Stage 1: learning subspace metrics

1 **for** each $\lambda \in \Lambda$ **do**

2     Recover $\hat{Q}_\lambda \in \mathrm{Sym}(\mathbb{R}^{r_\lambda})$ with respect to $B_\lambda$ via reduction to Algorithm 5 [Canal et al., 2022]

// Stage 2: reconstruction

3 Use ordinary least squares to solve the linear regression problem over $A \in \mathrm{Sym}(\mathbb{R}^d)$:

$$\hat{M}_{\mathrm{LS}} \leftarrow \operatorname*{argmin}_{A \in \mathrm{Sym}(\mathbb{R}^d)} \sum_{\lambda \in \Lambda} \left\| \hat{Q}_\lambda - B_\lambda^\top A B_\lambda \right\|_{\mathrm{F}}^2 \quad (4)$$

4 Project $\hat{M}_{\mathrm{LS}}$ onto the set of positive semidefinite $d \times d$ matrices by solving the convex optimization problem:

$$\hat{M} \leftarrow \operatorname*{argmin}_{A \succeq 0} \left\| A - \hat{M}_{\mathrm{LS}} \right\|_{\mathrm{F}}^2 \quad (5)$$

**Output:** $\hat{M}$.

**Divide-and-conquer algorithm** Algorithm 2 generalizes our earlier algorithm for unquantized measurements. As before, say we have obtained measurements for a set of items subspace-clusterable over $(V_\lambda)_\lambda$. In the first stage, we recover the subspace metrics on each $V_\lambda$. Lemma 8 reduces metric learning on subspaces to metric learning on $\mathbb{R}^r$, where $r$ is the dimension of the subspace, so we can call existing methods for metric learning from binary responses across users (Canal et al. [2022] or Algorithm 5). Thus, we obtain an estimator $\hat{Q}_\lambda$ for each subspace metric $Q_\lambda$.

In the second stage, we approximately reconstruct $M$ from the estimators $\hat{Q}_\lambda$. When each $\hat{Q}_\lambda$ was exact, we could just solve the linear system of equations $\Pi_{V_\lambda}(\hat{M}) = \hat{Q}_\lambda$. As this is no longer the case, we instead compute the ordinary least squares estimator $\hat{M}_{\mathrm{LS}}$, which minimizes $\sum_\lambda \| \hat{Q}_\lambda - \Pi_{V_\lambda}(A) \|^2$ over $A \in \mathrm{Sym}(\mathbb{R}^d)$ in Eq. (4) of Algorithm 2. Finally, we ensure that the reconstructed matrix corresponds to a pseudo-metric by solving a linear program to project $\hat{M}_{\mathrm{LS}}$ onto the cone of positive semi-definite matrices [Boyd and Vandenberghe, 2004].

## 5.1 RECOVERY GUARANTEES

**Reconstruction guarantee** The following theorem gives a recovery guarantee on the full metric, given approximate recovery for each subspace metric, $\| \hat{Q}_\lambda - Q_\lambda \|_{\mathrm{F}} \leq \varepsilon$ for some $\varepsilon > 0$. See Appendix E.1 for proof.

**Theorem 15.** *Let $\mathbb{R}^d$ have a Mahalanobis distance with matrix representation $M \in \mathrm{Sym}^+(\mathbb{R}^d)$. Let $\mathcal{X} \subset \mathbb{R}^d$ be subspace-clusterable over subspaces $V_\lambda$ indexed by $\lambda \in \Lambda$, where $|\Lambda| = n$. Let $\hat{M}$ be the estimator of $M$ and let $\hat{Q}_\lambda$ be the estimator of the subspace metric $Q_\lambda$ for each $\lambda$ learned*

*from Algorithm 2. Suppose there exist $\gamma \leq \varepsilon$ such that $\left\| \mathbb{E}[\hat{Q}_\lambda] - Q_\lambda \right\|_{\mathrm{F}} \leq \gamma$ and $\left\| \hat{Q}_\lambda - Q_\lambda \right\|_{\mathrm{F}} \leq \varepsilon$ for each $\lambda$. Fix $p \in (0, 1]$. Then, there is a universal constant $c > 0$ such that with probability at least $1 - p$,*

$$\left\| \hat{M} - M \right\|_{\mathrm{F}} \leq c \cdot \frac{1}{\sigma_{\min}(\Pi)} \left( \gamma \sqrt{n} + \varepsilon d \sqrt{\log \frac{2d}{p}} \right),$$

*where $\sigma_{\min} > 0$ is the least singular value of $\Pi$.*

**Remark 16.** *This recovery guarantee depends on three parameters: (1) $\sigma_{\min}(\Pi)$ captures how well-spread the set $\{xx^\top : x \in V_\lambda, \lambda \in \Lambda\}$ is across $\mathrm{Sym}(\mathbb{R}^d)$. (2) $\varepsilon$ bounds the recovery error for each subspace metric; it decreases as the number of pairwise comparisons per user increases (Remark 20). (3) $\gamma$ bounds the bias of the estimator $\hat{Q}_\lambda$. It can be the dominating term in the recovery bound, for example when $\sigma_{\min}(\Pi) \gg d$. While this bias term $\gamma \leq \varepsilon$ can be made arbitrarily small with enough comparisons per user, for data-starved regimes, bias reduction can also be applied in practice (e.g. Firth [1993]).*

**Recovery guarantee for subspace metrics** For completeness, we adapt the setting and results of [Canal et al., 2022] to provide a recovery guarantee for learning each subspace metric. We assume the same probabilistic model:

**Assumption 17** (Probabilistic model). *Let $M \in \mathrm{Sym}^+(\mathbb{R}^d)$ be the matrix representation of the Mahalanobis distance, let $v_1, \ldots, v_K \in \mathbb{R}^d$ be the pseudo-ideal points for a collection of users, and let $\mathcal{X} \subset \mathbb{R}^d$ be a set of items. We assume:*

$$\|M\|_{\mathrm{F}} \leq \zeta_M, \qquad \|v_k\| \leq \zeta_v, \qquad \sup_{x \in \mathcal{X}} \|x\| \leq 1,$$

*for $\zeta_M, \zeta_v > 0$. When asked to compare two items $x$ and $x'$, the $k$th user provides a binary response $Y$ with:*

$$\Pr[Y = y] = f\big(y \cdot \psi_M(x, x'; u_k)\big), \quad (6)$$

*where $f : \mathbb{R} \to [0, 1]$ is a strictly increasing link function such that $f(z) = 1 - f(-z)$, and where $u_k$ is the corresponding ideal point. On the domain $|z| \leq 2(\zeta_M + \zeta_v)$, let $f$ have lower bounded derivative $f'(z) \geq c_f$ and let the map $z \mapsto -\log f(z)$ have Lipschitz constant $L$.*

**Remark 18.** *The simple noise model on binary responses from Eq. (6) reflects human psychology [Coombs, 1964, Revelle, 2009]. When presented with two items to compare, our response is less noisy when a clear preference ranking exists. Conversely, when we are ambivalent between the two items, our response tends to be more random.*

Algorithm 5 estimates $(M, v_1, \ldots, v_K)$ by using the users' measurements to construct an optimization program over the parameters; when the loss function supplied to the algorithm is $\ell(z) = -\log f(z)$, the procedure is equivalent to maximum likelihood estimation. As noted above, it suffices to consider learning Mahalanobis distances on $\mathbb{R}^r$. The following proposition proves correctness of Algorithm 5.

**Proposition 19** (Theorem 4.1, Canal et al. [2022]). *Suppose that $\mathbb{R}^r$ has a Mahalanobis distance with representation $Q \in \mathrm{Sym}^+(\mathbb{R}^r)$ where $\|Q\|_{\mathrm{F}} \leq \zeta_M$. Let each user $k \in [K]$ have pseudo-ideal point $v_k \in \mathbb{R}^r$ where $v_k \leq \zeta_v$. Let $\mathcal{P}_m$ be a distribution over designs of size $m$ over $\mathbb{R}^r$ (Definition 1). For each user, let $D_k \sim \mathcal{P}_m$ be an i.i.d. random design, and let $\mathcal{D}_k = \{(x_{i_0}, x_{i_1}, y_{i;k})\}_{i \in [m]}$ be the user's responses under Assumption 17. Fix $p \in (0, 1]$. Given loss function $\ell(z) = -\log f(z)$, Algorithm 5 returns $\hat{Q} \in \mathrm{Sym}^+(\mathbb{R}^r)$, where with probability at least $1 - p$,*

$$\|\hat{Q} - Q\|_{\mathrm{F}}^2 \leq \frac{16L}{c_f^2 \cdot \sigma_{\min}^2(\mathcal{P}_m)} \sqrt{\frac{(\zeta_M^2 + K\zeta_v^2)\log\frac{4}{p}}{mK}}.$$

The proof of Proposition 19 is deferred to Appendix E.2.

**Remark 20.** *We can simplify the bound if we assume that $M$ has bounded entries, say $\|M\|_\infty \leq 1$. Let's also assume that user ideal points are contained in the unit ball, so that $\|u_k\|_2 \leq 1$ for each user. Then, we can set $\zeta_M \leq r$ and $\zeta_v \leq 2\sqrt{r}$ since $v_k = -2Mu_k$. Remark E.2 shows that given access to a subspace-clusterable set of items, we can construct a sequence of random designs $(\mathcal{P}_m)_m$ over those items such that $\sigma_{\min}^2(\mathcal{P}_m) = \Omega(1)$. Suppressing the confidence parameter $p$, we obtain the recovery guarantee:*

$$\|\hat{Q} - Q\|_{\mathrm{F}}^2 = \mathcal{O}\left(\sqrt{\frac{r^2 + Kr}{mK}}\right).$$

# 6 EMPIRICAL VALIDATION

In this section, we empirically validate our findings using synthetic data[3]. We aim to address the following questions:

1. Given limited noisy, quantized preference comparisons on subspace-clusterable items, can our proposed divide-and-conquer algorithm recover an unknown metric $M$?

2. Does the performance of our algorithm improve if we have access to more subspace-clusters, users or preference comparisons per user?

3. When items in $\mathcal{X}$ lie *approximately* in a union of subspaces, can we still recover $M$?

**Experimental setup** For each run, we generate a random ground-truth metric $M \in \mathrm{Sym}^+(\mathbb{R}^d)$ from the standard Wishart distribution $W(I_d, d)$, a collection of uniform-at-random $r$-dimensional subspaces [Stewart, 1980], and a set of user ideal points drawn i.i.d. from the Gaussian $\mathcal{N}(0, \frac{1}{d}I_d)$. Within each subspace, items are drawn i.i.d. from $\mathcal{N}(0, \frac{1}{r}BB^\top)$, where $B \in \mathbb{R}^{d \times r}$ is an orthonormal basis of that subspace. Given a user and a pair of items, a

---

[3]Our code is available at https://github.com/zhiwang123/metric-learning-lazy-crowds.

binary response is sampled according to the probabilistic model in Assumption 17,

$$\Pr[Y = y] = f(y \cdot \psi),$$

where $f$ is a link function. We select the logistic sigmoid link function, $f(z; \beta) = 1/(1 + \exp(-\beta z))$ in our experiments. By varying $\beta$, we can generate binary responses that interpolate between uncorrelated random noise ($\beta = 0$) and noiseless quantized measurments ($\beta = \infty$). Further details can be found in Appendix F.1.

We use our divide-and-conquer approach to learn the metric (Algorithm 2). To approximate the subspace metrics, this method calls Algorithm 5 (Stage 1, line 2). It does so by performing maximum likelihood estimation on the correctly-specified probabilistic model. When stitching the subspaces together, we observed that Huber regression [Huber, 1964, Pedregosa et al., 2011] generally leads to better performance over least squares regression within Algorithm 2 (Stage 2, line 3). In the following, we report results obtained using this robust variant of linear regression. We evaluate the learned metric $\hat{M}$ by its relative error, $\|\hat{M} - M\|_{\mathrm{F}}/\|M\|_{\mathrm{F}}$.

We ran three experiments each for 30 runs, where we set the subspace dimension to $r = 1$ and we set $\beta = 1$ in the logistic sigmoid link function.

**Experiment 1: Relative error vs number of comparisons** In the first experiment, we set the ambient dimension to $d = 10$ and generated data that lie in a union of 80 subspaces (by Remark 14, at least $\dim(\mathrm{Sym}(\mathbb{R}^{10})) = 55$ subspaces are needed for recovery). We ran Algorithm 2 for different combinations of $K$ and $m$, where $K$ is the number of users per subspace and $m$ is the number of preference comparisons per user. Figure 3a compares the average relative errors for varying $K$ and $m$. This experiment shows that with more preference comparisons, recovery within each subspace improves and we achieve better recovery of the full metric; this supports Theorem 15 and Proposition 19. This experiment also suggests that given 1-dimensional subspaces, even asking for only two measurements per user is sufficient to achieve good empirical performance for metric recovery.

**Experiment 2: Relative error vs number of subspaces** In the second experiment, we set $K = 60$ and $m = 4$. For ambient dimensions $d = 3, 4, \ldots, 10$, we consider the relative error for reconstructing $\hat{M}$ using an increasing number of subspaces, $n = 5, 6, \ldots, 80$. Figure 3b shows the average relative errors. For each $d$, average relative error decreases as $n$ increases. Furthermore, even in this non-idealized setting where users provide noisy, binary responses, we can obtain non-trivial relative error when the number of subspaces $n$ exceeds the information-theoretic bound $d(d + 1)/2$. This corroborates the dimension-counting argument in Remark 14 beyond unquantized measurements.

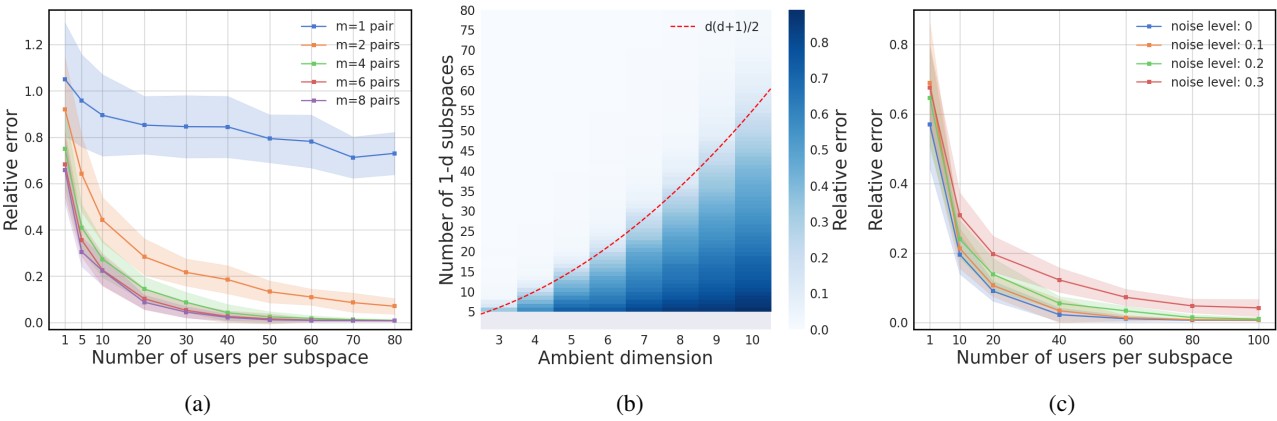

Figure 3: (a) shows the average relative errors for varying numbers of users per subspace and preference comparisons per user, where items lie in a union of 80 1-dimensional subspaces of $\mathbb{R}^{10}$. (b) shows the average relative errors given increasing numbers of 1-dimensional subspaces to reconstruct $\hat{M}$; for each subspace, 60 users each provides 4 preference comparisons. The dotted red curve illustrates the dimension-counting argument in Remark 14. (c) shows the average relative errors for varying subspace noise levels, where items lie approximately in a union of 80 1-dimensional subspaces of $\mathbb{R}^{10}$; each user provides 8 preference comparisons. The error bars in (a) and (c) represent one standard deviation from the mean.

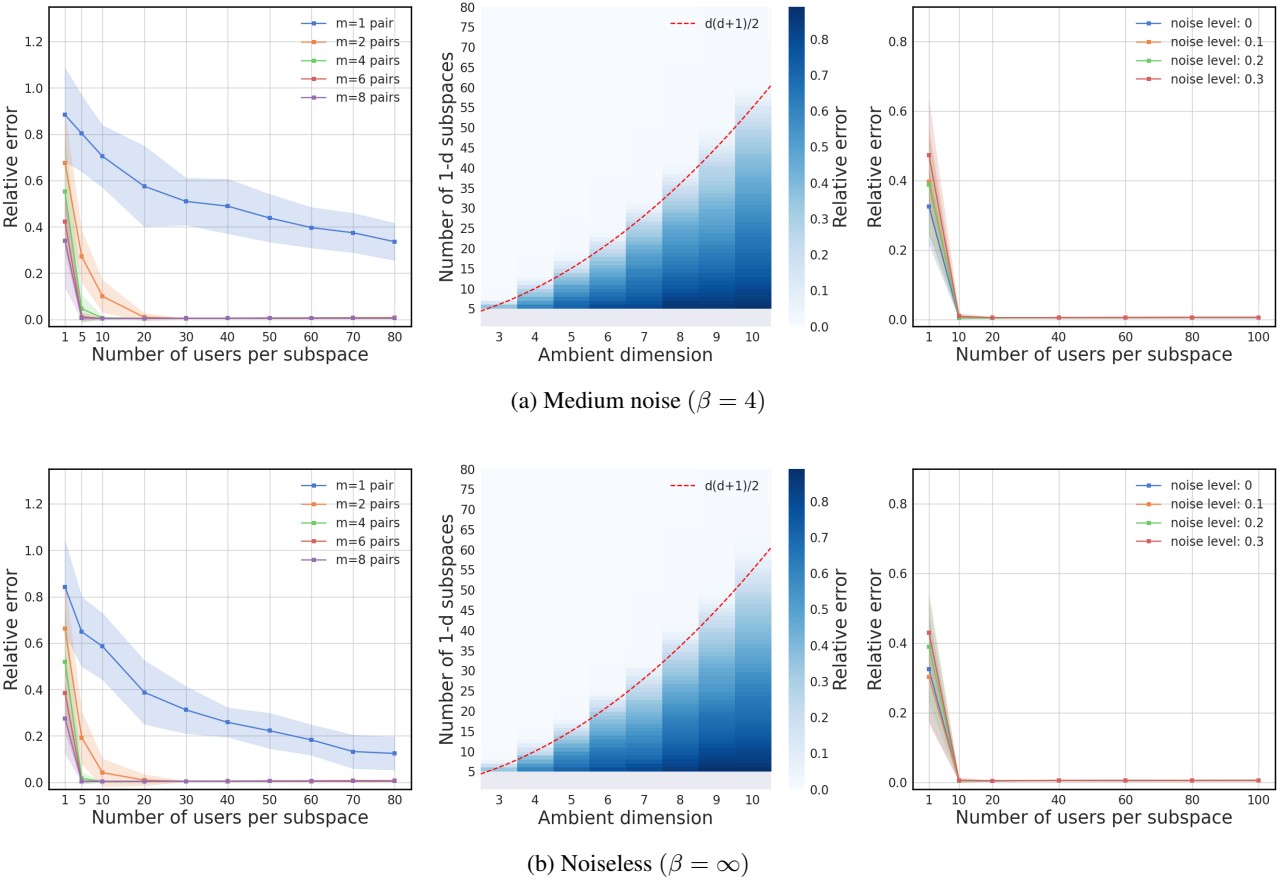

Figure 4: shows the results obtained from the three experiments with a misspecified response model. The learner is agnostic to the response noise level used to generate the data ($\beta = 4$ and $\beta = \infty$) and assumes $\beta = 1$ when recovering subspace metrics before stitching them together.

**Experiment 3: Recovery when items approximately lie in subspaces** In the third experiment, we empirically study how our approach works when the subspace clusterable assumption only approximately holds. For a subspace $V$, we sample items near $V$ from $\mathcal{N}(0, \frac{1}{r}BB^\top + \frac{\sigma^2}{d-r}B_\perp B_\perp^\top)$, where $\sigma > 0$ is a given noise level, $B \in \mathbb{R}^{d \times r}$ and $B_\perp \in \mathbb{R}^{d \times (d-r)}$ are orthonormal bases of $V$ and its orthogonal complement $V^\perp$, respectively. The way user preference responses are generated remains the same as before. For each subspace $V$, we preprocess the items by running singular value decomposition on the nearby items to recover an $r$-dimensional subspace $\hat{V}$. We project these items to $\hat{V}$, before running Algorithm 2 with these approximate representations. We set $d = 10$ and $m = 8$. For each subspace noise level $\sigma$, we ran our approach on items that lie approximately in 80 subspaces for varying $K$; Figure 3c shows the average relative errors. When the noise level $\sigma$ is low, we can still recover the metric well. This may break down as $\sigma$ increases; indeed, when $\sigma = 1$, there is no subspace structure at all.

**Learning with a misspecified response model** Recall that the subspace metrics are learned via maximum likelihood estimation (Algorithm 5, line 2). In this section, we investigate how sensitive this approach is to a misspecified response model. To do so, we repeated the above three experiments with a fixed model that assumes $\beta = 1$. However, we generated the data from mis-matched response noise levels: (a) $\beta = 4$, corresponding to the "medium" setting in [Canal et al., 2022], and (b) $\beta = \infty$, which is the noiseless setting where $y = -1$ if the corresponding unquantized measurement is less than 0 and $y = +1$ otherwise. Figure 4 shows that the learner performs well even without the correct knowledge of $\beta$. In Appendix F, we show that the approach still achieves reasonable performance when the negative log loss (to compute the maximum likelihood) is replaced with the hinge loss in Algorithm 5 (Figure F.1). This shows that the learner may not need to know that probabilistic model under which user binary responses are generated.

See also Appendix F.2 for additional experimental results with subspace dimension $r = 2$.

# 7 CONCLUSION AND FUTURE WORK

We studied crowd-based metric learning from very few preference comparisons per user. In general, we showed nothing can be learned. However, when the items exhibit low-rank subspace-clusterable structure, we proposed a divide-and-conquer approach and provided recovery guarantees. Interestingly, this work suggests that when training of foundation models, there is reason to favor learning general-purpose representations with low-rank structures, as this may reduce the cost of downstream fine-tuning and alignment.

Our experiments show that even when the items do not exactly lie on the subspaces, but instead only exhibit approximate subspace structure, our method can still recover the metric. We leave establishing theoretical gurantees for this setting for future work. Our results has implications for alignment of representations from foundation models to human preferences and we defer building an algorithmic framework that finds subspace clusters before learning metrics, and evaluating it with real-world item embeddings and human preference feedback for future work.

# 8 ACKNOWLEDGEMENTS

ZW thanks Kamalika Chaudhuri for helpful discussions and the National Science Foundation under IIS 1915734 for support. This work was also partially supported by NSF grants NCS-FO 2219903 and NSF CAREER Award CCF 2238876. Additionally, this work was partially supported by the NSF awards: SCALE MoDL-2134209, CCF-2112665 (TILOS). It was also supported in part by the DARPA AIE program, the U.S. Department of Energy, Office of Science, the Facebook Research Award, as well as CDC-RFA-FT-23-0069 from the CDC-s Center for Forecasting and Outbreak Analytics.

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

# Metric Learning from Limited Pairwise Preference Comparisons
## (Supplementary Material)

**Zhi Wang**[†1]          **Geelon So**[2]          **Ramya Korlakai Vinayak**[1]

[1] Dept. of Electrical and Computer Engineering,  University of Wisconsin-Madison
[2] Dept. of Computer Science and Engineering,  University of California San Diego

zhi.wang@wisc.edu, geelon@ucsd.edu, ramya@ece.wisc.edu

## OUTLINE OF THE SUPPLEMENTARY MATERIAL

---

[†]Work done at University of California San Diego.

# A  ADDITIONAL ALGORITHMS FROM EXISTING WORK

Algorithm 3 and Algorithm 4 describe the procedures for learning an unknown Mahalanobis distance using unquantized measurements from a single user and a large pool of users, respectively. See Section 2 of [Canal et al., 2022].

Algorithm 5 describes the convex optimization problem introduced in Section 3 of [Canal et al., 2022] for simultaneous metric and preference learning using quantized measurements from multiple users. Here, $\ell : \mathbb{R} \to \mathbb{R}_{0+}$ can be any convex loss function that is $L$-Lipschitz-continuous. In particular, to achieve the recovery guarantee in Proposition 19, we assume the probabilistic model in Assumption 17 with link function $f$ and use the loss function $\ell(z) = -\log f(z)$.

---

**Algorithm 3:** Metric learning using unquantized measurements from a single user [Canal et al., 2022]

---

**Input:** A set $\mathcal{D} = \left\{ (x_{i_0}, x_{i_1}, \psi_i) \right\}_{i=1}^m$ of unquantized measurements from a single user.

1 Solve the system of linear equations over symmetric matrices $A \in \mathbb{R}^{d \times d}$ and vectors $w \in \mathbb{R}^d$:

$$\left\langle x_{i_0} x_{i_0}^\top - x_{i_1} x_{i_1}^\top, A \right\rangle + \left\langle x_{i_0} - x_{i_1}, w \right\rangle = \psi_i.$$

**Output:** $\hat{A}$, the solution to the above linear equations.

---

**Algorithm 4:** Metric learning using unquantized measurements from multiple users [Canal et al., 2022]

---

**Input:** A family of $\mathcal{D}_k = \left\{ (x_{i_0;k}, x_{i_1;k}, \psi_{i;k}) \right\}_{i=1}^{m_k}$ of unquantized measurements from users $k \in [K]$.

1 Solve the system of linear equations over symmetric matrices $A \in \mathbb{R}^{d \times d}$ and vectors $w_1, w_2, \ldots, w_K \in \mathbb{R}^d$:

$$\left\langle x_{i_0;k} x_{i_0;k}^\top - x_{i_1;k} x_{i_1;k}^\top, A \right\rangle + \left\langle x_{i_0;k} - x_{i_1;k}, w_k \right\rangle = \psi_{i;k}.$$

**Output:** $\hat{A}$, the solution to the above linear equations.

---

**Algorithm 5:** Metric learning using quantized measurements from multiple users [Canal et al., 2022]

---

**Input:** A family of $\mathcal{D}_k = \left\{ (x_{i_0;k}, x_{i_1;k}, y_{i;k}) \right\}_{i=1}^{m_k}$ of quantized measurements from users $k \in [K]$; hyperparameters $\zeta_M, \zeta_v > 0$.

1 Solve the convex optimization problem over symmetric matrices $A \in \mathbb{R}^{d \times d}$ and vectors $w_1, w_2, \ldots w_K \in \mathbb{R}^d$:

$$\hat{M}, \{\hat{v}_k\}_k \leftarrow \min_{A, \{w_k\}_k} \quad \sum_k \sum_{\mathcal{D}_k} \ell \left( y_{i;k} \left( \left\langle x_{i_0} x_{i_0}^\top - x_{i_1;k} x_{i_1;k}^\top, A \right\rangle + \left\langle x_{i_0;k} - x_{i_1;k}, w_k \right\rangle \right) \right) \tag{7}$$
$$\text{s.t.} \quad A \succeq 0, \ \|A\|_{\mathrm{F}} \leq \zeta_M, \ \|w_k\|_2 \leq \zeta_v \ \forall k$$

**Output:** $\hat{M}$.

---

# B  DIRECT SUMS OF INNER PRODUCT SPACES

In the paper, we have liberally made use of direct sums of inner product spaces, for example, $\mathrm{Sym}(\mathbb{R}^d) \oplus \mathbb{R}^d$, which we treat as an inner product space. It allows us ready access to well-established machinery including inner products, norms, singular values, and pseudoinverses. The direct sum of inner product spaces is defined:

**Definition B.1.** *Let $\left( V, \langle \cdot, \cdot \rangle_V \right)$ and $\left( W, \langle \cdot, \cdot \rangle_W \right)$ be two inner product spaces. Their* direct sum *is the vector space $V \oplus W$ equipped with the inner product:*

$$\langle v_1 \oplus w_1, v_2 \oplus w_2 \rangle_{V \oplus W} = \langle v_1, v_2 \rangle_V + \langle w_1, w_2 \rangle_W.$$

*In particular, this induces the norm on $V \oplus W$ satisfying $\|v \oplus w\|_{V \oplus W}^2 = \|v\|_V^2 + \|w\|_W^2$.*

**Moore-Penrose pseudoinverse**    The pseudoinverse can be defined for any map between inner product spaces:

**Definition B.2.** *Let $A : V \to W$ be a linear map between inner product spaces $V$ and $W$. Let $K = \ker(A)$ and let $K^\perp$ be its orthogonal complement. Let $A_{K^\perp} : K^\perp \to \mathrm{Im}(A)$ be the restriction of $A$ to $K^\perp$ and let $\Pi_{\mathrm{Im}(A)} : W \to \mathrm{Im}(A)$ be the orthogonal projection onto $\mathrm{Im}(A)$. The* Moore-Penrose pseudoinverse *of $A$ is the map $A^+ : W \to V$ given by:*

$$A^+ = A_{K^\perp}^{-1} \circ \Pi_{\mathrm{Im}(A)}.$$

*Note that $A_{K^\perp}^{-1}$ exists by the first isomorphism theorem of algebra.*

**Universal property**    The following property of direct sum allows us to decompose a linear map $A : V_1 \oplus V_2 \to V$, which we use in the proof of Theorem 15, when decomposing $\Pi^+ : \bigoplus_\lambda \mathrm{Sym}(V_\lambda) \to \mathrm{Sym}(\mathbb{R}^d)$.

**Proposition B.3** (Universal property of the direct sum, Mac Lane and Birkhoff [1999]). *Let $A : V_1 \oplus V_2 \to V$ be a linear map. Then, there exists $A_i : V_i \to V$ for $i = 1, 2$ such that for all $v_1 \oplus v_2 \in V_1 \oplus V_2$,*

$$A(v_1 \oplus v_2) = A_1(v_1) + A_2(v_2).$$

**Schatten norm**    The Frobenius norm over matrices can be generalized to linear maps between inner product spaces:

**Definition B.4.** *Let $A : V \to W$ be a linear map between finite-dimensional inner product spaces of rank $r$. Let $\sigma_1 \geq \cdots \geq \sigma_r$ be its nonzero singular values. The* 2-Schatten norm *$\|A\|_2$ is given by:*

$$\|A\|_2^2 = \sum_{i=1}^r \sigma_i^2.$$

*In particular, this implies $\|A\|_2 \leq \sigma_{\max}(A) \cdot \sqrt{\mathrm{rank}(A)}$.*

**Proposition B.5.** *Let $A : V_1 \oplus V_2 \to V$ be a linear map between finite-dimensional inner product spaces. Let $A_i : V_i \to V$ for $i = 1, 2$ be given as in Proposition B.3. Then:*

$$\|A\|_2^2 = \|A_1\|_2^2 + \|A_2\|_2^2,$$

*where $\| \cdot \|_2$ denotes the 2-Schatten norm.*

## C    PROOFS AND ADDITIONAL RESULTS FOR SECTION 3

For the proof of Theorem 3, we will make use of the notion of a *comparison graph* over a set of items. Given preference comparisons from a user, the induced comparison graph is simply the directed graph over items where two items are connected by an edge if the user has compared them:

**Definition C.1.** *A comparison graph $G = (V, E)$ is a graph whose vertices $V = \{x_1, \ldots, x_N\}$ is a set of items and whose edges $E = \{(x_{i_0}, x_{i_1})\}_{i=1}^m$ is a set of item pairs. Its* edge-vertex incidence matrix *$S \in \{-1, 0, +1\}^{m \times N}$ is defined by:*

$$S_{ij} = \begin{cases} 1 & j = i_0 \\ -1 & j = i_1 \\ 0 & o.w. \end{cases}$$

**Theorem 3.** *Fix $M \in \mathrm{Sym}^+(\mathbb{R}^d)$ and $v_k \in \mathbb{R}^d$ for each $k \in \mathbb{N}$. Let $(D_k)_{k \in \mathbb{N}}$ be a collection of design matrices, each for a set of $m \leq d$ pairwise comparisons. If each set of compared items has generic pairwise relations, then for all $M' \in \mathrm{Sym}^+(\mathbb{R}^d)$, there exists $(v_k')_{k \in \mathbb{N}} \subset \mathbb{R}^d$ such that:*

$$D_k(M, v_k) = D_k(M', v_k'), \qquad \forall k \in \mathbb{N}.$$

*Proof of Theorem 3.* Fix $M' \in \mathrm{Sym}^+(\mathbb{R}^d)$. It suffices to prove the result for a single user, since the covariates $v_k$'s impose no constraints on each other. Fix a pseudo-ideal point $v \in \mathbb{R}^d$. Let $D$ be a design matrix induced by the collection of pairs $\{(x_{i_0}, x_{i_1})\}_{i=1}^m$ from a set of items $\mathcal{X} = \{x_1, \ldots, x_N\}$. We show that when $\mathcal{X}$ has generic pairwise relations, then there exists $v' \in \mathbb{R}^d$ such $D(M, v) = D(M', v')$. By expanding and rearranging this equation, we obtain a linear system of equations $Av' = b$, where $A \in \mathbb{R}^{m \times d}$ and $b \in \mathbb{R}^m$, and where the $i$th set of equations is given by:

$$\underbrace{(x_{i_0} - x_{i_1})^\top}_{i\text{th row of } A} v' = \underbrace{\left\langle x_{i_0} x_{i_0}^\top - x_{i_1} x_{i_1}^\top, M - M' \right\rangle + \left\langle x_{i_0} - x_{i_1}, v \right\rangle}_{i\text{th entry of } b}.$$

The Rouché-Capelli theorem states that the system $A\hat{u} = b$ has a solution if the rank of the augmented matrix $[A|b]$ is equal to the rank of the design matrix $A$. If this is the case, then, there is a solution $v'$ for any choice of $M' \in \mathrm{Sym}^+(\mathbb{R}^d)$.

To finish the proof, we show that the ranks of $A$ and $[A|b]$ are equal. To this end, let $S$ be the edge-vertex incidence matrix induced by $\{(x_{i_0}, x_{i_1})\}_{i=1}^m$. Define the matrix $X \in \mathbb{R}^{N \times d}$ and vector $b' \in \mathbb{R}^N$ so that the $j$th row of each is:

$$X_j = x_j^\top \qquad \text{and} \qquad b_j' = \left\langle x_j x_j^\top, M - M' \right\rangle + \left\langle x_j - v \right\rangle,$$

so that $A = SX$ and $b = Sb'$. The items have generic pairwise relations, and so $\mathrm{rank}(S) = \mathrm{rank}(SX)$ by Lemma C.2. The augmented matrix $[A|b]$ has the decomposition $S[X|b']$, so its rank is upper bounded by $\mathrm{rank}(S)$. And because the $\mathrm{rank}([A|b])$ is at least $\mathrm{rank}(A)$, we obtain equality, as claimed. $\square$

**Lemma C.2.** *Let $V = \{x_1, \ldots, x_N\}$ be a set of items in $\mathbb{R}^d$, and let $X \in \mathbb{R}^{N \times d}$ be its matrix representation, so that the $j$th row is $X_j = x_j^\top$. Let $G = (V, E)$ be a comparison graph with $|E| \leq d$. Let $S$ be its edge-vertex incidence matrix. If the items have generic pairwise relations, then:*

$$\mathrm{rank}(SX) = \mathrm{rank}(S).$$

*Proof.* Let $G' = (V, E')$ be a maximal acyclic subgraph $G' \subset G$, say with $m'$ edges, and let $S' \in \mathbb{R}^{m' \times N}$ be its corresponding edge-vertex incidence matrix. On the one hand, we have:

$$\mathrm{rank}(S') \geq \mathrm{rank}(S'X).$$

On the other, because $\mathcal{X}$ has pairwise generic relations and $m' \leq d$, we have:

$$\mathrm{rank}(S'X) = \dim\bigl(\mathrm{span}\bigl(\{x - x' : (x, x') \in E'\}\bigr)\bigr) = m' \geq \mathrm{rank}(S').$$

The first equality is obtained by the definition of rank applied to $S'X$. The second equality follows from pairwise genericity. Thus, we have $\mathrm{rank}(S') = \mathrm{rank}(S'X) \leq \mathrm{rank}(SX)$. Furthermore, we claim that:

$$\mathrm{rank}(S) = \mathrm{rank}(S').$$

It would follow that $\mathrm{rank}(S) \leq \mathrm{rank}(SX) \leq \mathrm{rank}(S)$, which implies the result.

We prove the claim by showing that for any $e \in E \setminus E'$, the row $S_e$ is a linear combination of rows $S_{e'}$ where $e' \in E'$. Let $e = (x, x')$. By the maximality of $G'$, a cycle containing $e$ is created by including $e$ into $G'$. Thus, there is an undirected path $P$ from $x$ to $x'$ in $G'$, where $P = (x_0, \ldots, x_k)$ satisfies:

- $x_0 = x$ and $x_k = x'$,
- either $(x_{i-1}, x_i)$ or its reversal $(x_i, x_{i-1})$ is contained in $E'$.

For each $i$, let $e_i \in E'$ be one of these edges $(x_{i-1}, x_i)$ or $(x_i, x_{i-1})$ and let $r_i \in \{-1, +1\}$ indicate whether $e_i$ was the reversal of $(x_{i-1}, x_i)$. It follows that indeed $S_e$ is a linear combination of the rows of $S'$,

$$S_e = \sum_{i=1}^k r_i S_{e_i}.$$

$\square$

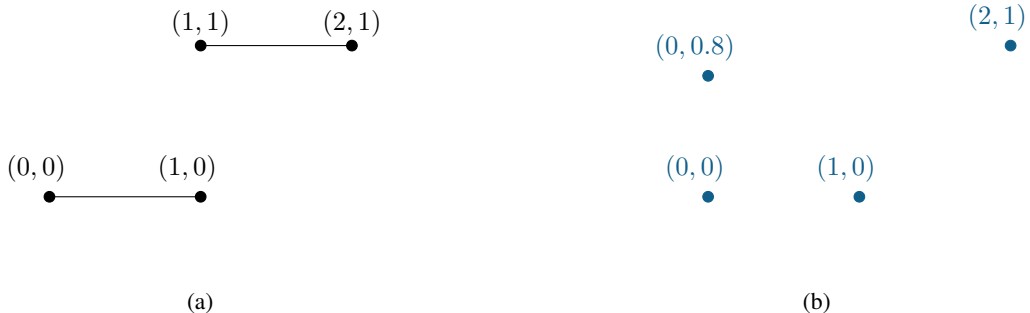

Figure C.1: (a) Illustration of Example C.6. The set of four points is in general linear position, but does not have generic pairwise relations. (b) A set of four points that has generic pairwise relations; it must also be in general linear position.

## C.1   GENERIC PAIRWISE RELATIONS

In the next proposition, we show that our notion of *generic pairwise relations* is a notion of points being in general position [Matousek, 2013]; almost all finite subsets of $\mathbb{R}^d$ have pairwise generic relations. Recall:

**Definition 2.** *A set $\mathcal{X} \subset \mathbb{R}^d$ has* generic pairwise relations *if for any acyclic graph $G = (\mathcal{X}, E)$ with at most $d$ edges, the set $\{x - x' : (x, x') \in E\}$ is linearly independent.*

**Proposition C.3.** *Fix $N \in \mathbb{N}$. We say that $X \in \mathbb{R}^{N \times d}$ has generic pairwise relations if its rows have generic pairwise relations. The following set has Lesbegue measure zero:*

$$\{X \in \mathbb{R}^{N \times d} : X \text{ is not pairwise generic}\}.$$

*Proof.* Let $\mathcal{S}$ be the finite collection of all edge-vertex incidence matrices $S$ for acyclic comparison graphs with at most $d$ edges on $N$ items. Notice that if $X \in \mathbb{R}^{N \times d}$ is not pairwise generic, then there exists some $S \in \mathcal{S}$ such that $SX \in \mathbb{R}^{m \times d}$ is not full rank. It follows that:

$$\{X \text{ not pairwise generic}\} = \bigcup_{S \in \mathcal{S}} \{\det(SXX^\top S^\top) = 0\}.$$

The zero set $\{\det(SXX^\top S^\top) = 0\}$ of a non-zero polynomial has Lebesgue measure zero, by Sard's theorem. The finite union of measure zero sets also has measure zero. $\qquad\square$

The concept of *general linear position* is a standard notion of general position. We present the definition in a way to highlight its relationship to pairwise genericity. Recall that a star graph is a tree with a root vertex connected to all other vertices.

**Definition C.4.** *Let $\mathcal{X}$ be a subset of $\mathbb{R}^d$. We say that $\mathcal{X}$ is in* general linear position *if for any star graph $G = (V, E)$ with at most $d$ edges on $V \subset \mathcal{X}$, the set $\{x - x' : (x, x') \in E\}$ is linearly independent.*

Because star graphs are acyclic graphs, the following is immediate:

**Proposition C.5.** *If $\mathcal{X}$ has generic pairwise relations, then $\mathcal{X}$ is in general linear position.*

On the other hand, the converse is not necessarily true. As we can see from the following example, having pairwise generic relations is a strictly stronger condition than being in general linear position; see also Figure C.1 for an illustration.

**Example C.6.** *Consider the following points in $\mathbb{R}^2$:*

$$x_1 = \begin{bmatrix} 0 \\ 0 \end{bmatrix} \qquad x_2 = \begin{bmatrix} 1 \\ 0 \end{bmatrix} \qquad x_3 = \begin{bmatrix} 1 \\ 1 \end{bmatrix} \qquad x_4 = \begin{bmatrix} 2 \\ 1 \end{bmatrix}.$$

*This collection of points is in general linear position, since no three points are collinear. However, these points do not have generic pairwise relations. We have:*

$$x_2 - x_1 = x_4 - x_3.$$

# D   PROOFS AND ADDITIONAL RESULTS FOR SECTION 4

## D.1   AN ADDITIONAL RESULT FOR SECTION 4.1

**Proposition D.1.** *There is a one-to-one correspondence between* $\mathrm{Sym}^+(V)$ *and Mahalanobis distances on* $V$*. In particular,* $\rho_V : V \times V \to \mathbb{R}$ *is a Mahalanobis distance if and only if there exists some* $Q \in \mathrm{Sym}^+(V)$ *such that:*

$$\rho_V(x, x') = \sqrt{(x - x')BQB^\top(x - x')}.$$

*Moreover, $Q$ is unique. We say that $Q$ is the* matrix representation *of the Mahalanobis distance $\rho_V$. If $\rho_V$ is the subspace metric on $V$ of a Mahalanobis distance $\rho$ on $\mathbb{R}^d$ with representation $M \in \mathrm{Sym}^+(\mathbb{R}^d)$, then:*

$$Q = \Pi_V(M).$$

*Proof.* ($\Longrightarrow$). Suppose that $\rho_V$ is a Mahalanobis distance on $V$. We show that it has a representation in $\mathrm{Sym}^+(V)$. By definition, there exists a Mahalanobis distance $\rho$ on $\mathbb{R}^d$ such that:

$$\rho_V = \rho\big|_V.$$

Let $M$ be the matrix representation of $\rho$ and let $Q = \Pi_V(M) \in \mathrm{Sym}^+(V)$. Then:

$$
\begin{aligned}
\rho_V(x, x') &= \sqrt{(x - x')^\top M(x - x')} \\
&= \sqrt{(x - x')^\top BB^\top MBB^\top(x - x')} \\
&= \sqrt{(x - x')^\top BQB^\top(x - x')},
\end{aligned}
$$

where the first equality expands the equality $\rho_V(x, x') = \rho(x, x')$, the second uses the fact that $BB^\top x = x$ for all $x \in V$ since $B \in \mathbb{R}^{d \times r}$ is an orthonormal basis, and the third equality is uses the definition of $\Pi_V$.

To prove uniqueness, suppose that $Q, Q' \in \mathrm{Sym}^+(V)$ represent $\rho_V$. We claim that $Q = Q'$. To show this, it suffices to prove that for all $z \in \mathbb{R}^r$,

$$\langle Q - Q', zz^\top \rangle = 0.$$

This is because the collection $\{zz^\top : z \in \mathbb{R}^r\}$ spans all $(r \times r)$-symmetric matrices. To this end, fix $z \in \mathbb{R}^r$. We take $x, x' \in V \subset$ by setting $x = Bz$ and $x' = 0$. We have:

$$\rho_V(x, x') = \sqrt{(x - x')^\top BQB^\top(x - x')} = \sqrt{z^\top Qz}.$$

The same equation holds for $Q'$ since both represent $\rho_V$. Squaring both equations and taking their difference shows that $\langle Q - Q', zz^\top \rangle = 0$, as desired. Thus, $Q = Q'$ and the matrix representation of $\rho_V$ is unique.

($\Longleftarrow$). Let $Q \in \mathrm{Sym}^+(V)$. We can extend the orthonormal basis $B$ of $V$ to an orthonormal basis of $\mathbb{R}^d$. In particular, let $B_\perp \in \mathbb{R}^{d \times (d-r)}$ be an orthonormal basis of the orthogonal complement of $V$. Set:

$$M = B_\perp B_\perp^\top + BQB^\top,$$

so that $M \in \mathrm{Sym}^+(\mathbb{R}^d)$ is positive-definite. Let $\rho$ be the Mahalanobis distance on $\mathbb{R}^d$ represented by $M$. Then, the Mahalanobis distance $\rho\big|_V$ on $V$ has representation:

$$\Pi_V(M) = B^\top MB = B^\top B_\perp B_\perp^\top B + B^\top BQB^\top B = Q,$$

which shows that each $Q \in \mathrm{Sym}^+(\mathbb{R}^d)$ corresponds to a Mahalanobis distance on $V$. $\qquad\square$

## D.2 PROOFS FOR SECTION 4.2

**Lemma 8.** *Let $V$ be an $r$-dimensional subspace of $\mathbb{R}^d$ with a canonical representation given by $B \in \mathbb{R}^{d \times r}$. Fix any Mahalanobis distance $M \in \mathrm{Sym}^+(\mathbb{R}^d)$, any pair of items $x, x' \in \mathbb{R}^d$, and ideal point $u \in \mathbb{R}^d$. Suppose that $x$ and $x'$ are contained in $V$ with canonical representation $x_V = B^\top x$ and $x'_V = B^\top x'$ in $\mathbb{R}^r$. Then:*

$$\psi_M(x, x'; u) = \psi_Q(x_V, x'_V; u_V),$$

*where the phantom ideal point $u_V$ of $u$ on $V$ satisfies $(B^\top M B)u_V = B^\top M u$, and $Q = \Pi_V(M)$ is the matrix representation in $\mathrm{Sym}^+(V)$ of the subspace metric $\rho\big|_V$.*

*Proof.* Let $v = -2Mu$ and $v_V = -2Qu_V$ be the pseudo-ideal user points for $u$ and $u_V$, respectively. The following shows that $v_V$ is given by the canonical representation of the orthogonal projection of $v$ to $V$,

$$v_V = -2 \underbrace{B^\top M B}_{Q} \underbrace{(B^\top M B)^{-1} B^\top M u}_{u_V} = -2 B^\top M u = B^\top v.$$

We now expand the definitions of $\psi_M$ and $\psi_Q$,

$$\psi_M(x, x'; u) \overset{(i)}{=} \left\langle xx^\top - x'x'^\top, M \right\rangle + \left\langle x - x', v \right\rangle$$
$$\overset{(ii)}{=} \left\langle BB^\top(xx^\top - x'x'^\top)BB^\top, M \right\rangle + \left\langle BB^\top(x - x'), v \right\rangle$$
$$\overset{(iii)}{=} \left\langle B^\top xx^\top B - B^\top x'x'^\top B, B^\top M B \right\rangle + \left\langle B^\top x - B^\top x', B^\top v \right\rangle$$
$$\overset{(iv)}{=} \left\langle x_V x_V^\top - x'_V x'^\top_V, Q \right\rangle + \left\langle x_V - x'_V, v_V \right\rangle$$
$$\overset{(v)}{=} \psi_Q(x_V, x'_V; u_V),$$

where (i) and (v) follow by definition, (ii) uses the fact that as $B \in \mathbb{R}^{d \times r}$ is an orthonormal basis, $BB^\top v = v$ for all $v \in V$, (iii) applies the following property for the trace inner product $\langle BA, C \rangle = \mathrm{tr}(C^\top BA) = \langle A, B^\top C \rangle$, and (iv) rewrites the equation in terms of the canonical representations. $\qquad \square$

**Proposition 10.** *Let $\mathcal{X}$ quadratically span a subspace $V$ of dimension $r$. There exists a collection $D_1, \dots, D_K$ of design matrices, each over $m$ pairs of items in $\mathcal{X}$, such that given a (distinct) user's response to each design, $\rho\big|_V$ can be identified when $m \geq r + 1$ and $K \geq r(r+1)/2$.*

*Proof.* By Lemma 8, it suffices to prove the result for $V = \mathbb{R}^d$. We show that if $\mathcal{X}$ quadratically spans $\mathbb{R}^d$, then we can construct an $(m, K)$-experimental design where $m = d + 1$ and $K = d(d+1)/2$ such that there is a unique matrix consistent with all user responses. Let $D = d + \frac{d(d+1)}{2}$ be the dimension of $\mathrm{Sym}(\mathbb{R}^d) \oplus \mathbb{R}^d$.

Since $\mathcal{X}$ quadratically spans $V$, there exists a collection of pairs $\{(x_{i_0}, x_{i_1})\}_{i=1}^D$ such that:

$$\mathrm{span}\left(\left\{ \Delta_i \oplus \delta_i : i \in [D] \right\}\right) = \mathrm{Sym}(\mathbb{R}^d) \oplus \mathbb{R}^d, \tag{8}$$

where we let $\Delta_i = x_{i_0} x_{i_0}^\top - x_{i_1} x_{i_1}^\top$ and $\delta_i = x_{i_0} - x_{i_1}$. In particular, the collection $\{\Delta_i \oplus \delta_i\}_{i \in [D]}$ is linearly independent. Without loss of generality, we may select these so that the first $d$ pairwise differences $\delta_i$ are also linearly independent:

$$\mathrm{span}\left(\left\{ \delta_i : i \in [d] \right\}\right) = \mathbb{R}^d.$$

We will ask all users to compare the first $d$ pairs and one additional pair, unique to the user. In particular, set the $k$th collection of preference comparison queries by:

$$\mathcal{D}_k = \left\{ (x_{i_0}, x_{i_1}) : i \in \mathcal{I}_k \right\}, \qquad \text{where } \mathcal{I}_k = [d] \cup \{d + k\}.$$

First, we show that the responses from a single user must reveal at least one dimension of $\mathrm{Sym}(\mathbb{R}^d)$. To see this, let's fix a user $k \in [K]$. From Equation (8), we can define the vector $(\alpha_{i,k} : i \in \mathcal{I}_k)$ so that:

$$\alpha_{d+k;k} = 1 \qquad \text{and} \qquad \sum_{i \in \mathcal{I}_k} \alpha_{i;k} \delta_i = 0.$$

Therefore, from the preference measurements, we deduce that at least one degree of freedom of $M$ is revealed:

$$\sum_{i\in\mathcal{I}_k}\alpha_{i;k}\psi_{i;k} = \sum_{i\in\mathcal{I}_k}\alpha_{i;k}\langle\Delta_i, M\rangle + \underbrace{\sum_{i\in\mathcal{I}_k}\alpha_{i;k}\langle\delta_i, v_k\rangle}_{\langle 0, v_k\rangle} = \left\langle\sum_{i\in\mathcal{I}_k}\alpha_{i;k}\Delta_i, M\right\rangle. \tag{9}$$

We now claim that each user reveals a different degree of freedom of $M$. In particular, it suffices to show that the following collection of matrices spans $\mathrm{Sym}(\mathbb{R}^d)$,

$$\left\{\sum_{i\in\mathcal{I}_k}\alpha_{i;k}\Delta_i : k\in[K]\right\}.$$

Suppose otherwise. Since $K = \frac{d(d+1)}{2}$, this means that this collection of matrices are linearly dependent, and that there exists a non-zero vector $(\mu_k : k\in[K])$ such that $0\in\mathrm{Sym}(\mathbb{R}^d)$ is the linear combination:

$$\sum_{k\in[K]}\mu_k\sum_{i\in\mathcal{I}_k}\alpha_{i;k}\Delta_i = 0.$$

Because we chose $\alpha_{d+k;k} = 1$ for each user $k\in[K]$, this implies that zero in $\mathrm{Sym}(\mathbb{R}^d)\oplus\mathbb{R}^d$ is also a non-trivial linear combination of the collection $\Delta_i\oplus\delta_i$, where:

$$\sum_{i=1}^{d}\left(\sum_{k\in[K]}\mu_k\alpha_{i;k}\right)\Delta_i\oplus\delta_i + \sum_{i=d+1}^{D}\mu_{i-d}\cdot\Delta_i\oplus\delta_i = 0.$$

But then this collection is not full rank and cannot span $\mathrm{Sym}(\mathbb{R}^d)\oplus\mathbb{R}^d$, as assumed in Equation (8). It follows that $M$ is the unique solution to the system of linear equations corresponding to Equation (9). □

**Proposition 11.** *Let $(D_k)_{k\in\mathbb{N}}$ be a set of design matrices over items in $\mathcal{X}\subset V$. If $\mathcal{X}$ does not quadratically span $V$, then infinitely many Mahalanobis distances on $V$ are consistent with any set of user responses to the design matrices.*

*Proof.* Because $\mathcal{X}_V$ does not quadratically span $V$, there exists an element $Q_\perp\oplus v_\perp\in\mathrm{Sym}(V)\oplus V$ such that:

$$\left\langle\left(x_V x_V^\top - x_V' x_V'^\top\right)\oplus(x-x'), Q_\perp\oplus v_\perp\right\rangle = 0,$$

for all $x, x'\in\mathcal{X}_V$, where $x_V = B^\top x$ and $x_V' = B^\top x'$. Let $M_\perp = BQB^\top$, so that:

$$\left\langle\left(xx^\top - x'x'^\top\right)\oplus(x-x'), M_\perp\oplus v_\perp\right\rangle = 0, \qquad\text{for all } x, x'\in\mathcal{X}_V. \tag{10}$$

We claim that if $M\in\mathrm{Sym}^+(\mathbb{R}^d)$ is consistent with the $k$th user's responses $\mathcal{D}_k = \{(x_{i_0;k}, x_{i_1;k}, \psi_{i;k})\}_{i=1}^m$, then the matrix $M+\lambda M_\perp$ is also consistent, provided that $M+\lambda M_\perp$ remains in $\mathrm{Sym}^+(\mathbb{R}^d)$. In particular, if $M$ is consistent, there exists an ideal point $u_k$ so that for all $i\in[m]$:

$$\begin{aligned}\psi_{i;k} = \psi_M(x_{i_0}, x_{i_1}; u_k) &\stackrel{(i)}{=} \left\langle\left(x_{i_0}x_{i_0}^\top - x_{i_1}x_{i_1}^\top\right)\oplus(x_{i_0}-x_{i_1}), M\oplus v_k\right\rangle\\
&\stackrel{(ii)}{=} \left\langle\left(x_{i_0}x_{i_0}^\top - x_{i_1}x_{i_1}^\top\right)\oplus(x_{i_0}-x_{i_1}), M\oplus v_k + \lambda M_\perp\oplus v_\perp\right\rangle\\
&\stackrel{(iii)}{=} \psi_{M+\lambda M_\perp}(x_{i_0}, x_{i_1}; \lambda\tilde{u}_k),\end{aligned}$$

where (i) expands the definition of $\psi_M$ while setting the pseudo-ideal point to $v_k = -2Mu_k$, (ii) applies Equation (10), and (iii) applies the definition of $\psi_{M+M_\perp}$ while setting $\tilde{u}_k = -\frac{1}{2}M^{-1}(v_k+v_\perp)$.

Thus, if $M$ is the matrix representation of the underlying Mahalanobis distance, the following matrices are also consistent:

$$\left\{M+\lambda M_\perp : 0\le\lambda < \frac{\sigma_{\min}(M)}{\sigma_{\max}(M_\perp)}\right\},$$

where $\sigma_{\max}(M_\perp)$ is the maximum singular value of $M_\perp$ while $\sigma_{\min}(M)$ is the minimum singular value of $M$; this implies that $M+\lambda M_\perp$ is positive-definite. Infinitely such $\lambda$'s exist because (a) $\sigma_{\max}(M_\perp) < \infty$ is finite and (b) $\sigma_{\min}(M) > 0$ is bounded away from zero because $M$ is positive-definite. □

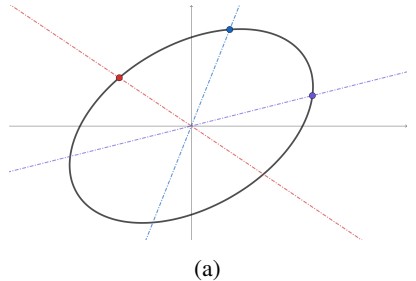 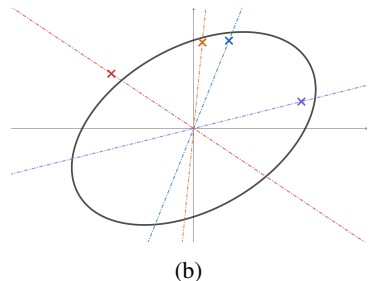

(a)                                     (b)

Figure D.1: (a) Illustrates the number of subspaces needed to reconstruct a high-dimensional ellipsoid from its intersections with low-dimensional subspaces. In $\mathbb{R}^2$, we need 3 points on distinct 1-dimensional subspaces to possibly recover an ellipse centered at the origin. (b) When we cannot exactly identify where the high-dimensional ellipsoid intersects with each subspace, we may still fit an ellipsoid from approximate estimations using least squares [Gander et al., 1994].

### D.3    PROOF OF PROPOSITION 13 FROM SECTION 4.3

**Proposition 13.** *Let $\rho$ be a Mahalanobis distance on $\mathbb{R}^d$. Let $(V_\lambda)_{\lambda \in \Lambda}$ be a collection of subspaces with canonical representations given by the orthonormal bases $(B_\lambda)_{\lambda \in \Lambda}$. The following are equivalent:*

1. *$\left\{ xx^\top : x \in V_\lambda, \lambda \in \Lambda \right\}$ spans $\mathrm{Sym}(\mathbb{R}^d)$.*

2. *Let $\Pi_{V_\lambda}$ be given by Equation (3). The linear map $\Pi : \mathrm{Sym}(\mathbb{R}^d) \to \bigoplus_{\lambda \in \Lambda} \mathrm{Sym}(V_\lambda)$ is injective, where:*

$$\Pi(A) = \bigoplus_{\lambda \in \Lambda} \Pi_{V_\lambda}(A).$$

3. *If $\hat{\rho}$ is a Mahalanobis distance such that $\hat{\rho}\big|_{V_\lambda} = \rho\big|_{V_\lambda}$ for all $\lambda \in \Lambda$, then $\hat{\rho} = \rho$.*

*Proof.* $(1 \implies 2)$. Suppose $\mathrm{span} \left\{ xx^\top : x \in V_\lambda, \lambda \in \Lambda \right\} = \mathrm{Sym}(\mathbb{R}^d)$. To show that $\Pi$ is injective, it suffices to show that its kernel is trivial. Let $M \in \ker(\Pi)$. We claim that for any $\lambda \in \Lambda$ and $x \in V_\lambda$, we have:

$$\left\langle xx^\top, M \right\rangle = 0. \tag{11}$$

Assume this for now. Then, $M \in \mathrm{Sym}(\mathbb{R}^d) = \mathrm{span} \left\{ xx^\top : x \in V_\lambda, \lambda \in \Lambda \right\}$, so that $\langle M, M \rangle = 0$. This implies that $M = 0$, so the kernel is trivial. We now show Eq. (11). Using the definition of $\Pi_{V_\lambda}$, when $M \in \ker(\Pi)$, we have:

$$\Pi_{V_\lambda}(M) = B_\lambda^\top M B_\lambda = 0. \tag{12}$$

Say that $\dim(V_\lambda) = r_\lambda$ and $x \in V_\lambda$. As $B_\lambda \in \mathbb{R}^{d \times r_\lambda}$ is a basis of $V_\lambda$, there exists $z \in \mathbb{R}^{r_\lambda}$ such that $x = B_\lambda z$. By Eq. (12),

$$\left\langle xx^\top, M \right\rangle = z^\top B_\lambda^\top M B_\lambda z = 0.$$

$(2 \implies 1)$. We prove the contrapositive. Suppose that $S = \mathrm{span} \left\{ xx^\top : x \in V_\lambda, \lambda \in \Lambda \right\}$ does not span $\mathrm{Sym}(\mathbb{R}^d)$. Then, there exists some nonzero $A \in S^\perp$ in its orthogonal complement. To show that $\Pi$ is not injective, we show that $A \in \ker(\Pi)$. That is, for all $\lambda \in \Lambda$, that $B_\lambda^\top A B_\lambda = 0$. We do this by proving that all eigenvalues of $B_\lambda^\top A B_\lambda$ are zero.

Let $v \in \mathbb{R}^{r_\lambda}$ be any unit eigenvector of $B_\lambda^\top A B_\lambda$ and $\alpha$ be the corresponding eigenvalue, so that:

$$\alpha = v^\top B_\lambda^\top A B_\lambda v = \left\langle xx^\top, A \right\rangle,$$

where $x = B_\lambda v$ is an element of $V_\lambda$. But because $A \in S^\perp$, this implies that the eigenvalue is zero, $\alpha = 0$.

$(2 \implies 3)$. Let $M$ and $\hat{M}$ be the matrix representations of $\rho$ and $\hat{\rho}$, respectively. By assumption, their subspace metrics coincide over $(V_\lambda)_\lambda$, so Proposition D.1 implies:

$$\Pi_{V_\lambda}(M) = \Pi_{V_\lambda}(\hat{M}).$$

And as $\Pi$ is injective, we must have $M = \hat{M}$, so that $\rho = \hat{\rho}$.

$(3 \implies 2)$. We prove that $\Pi$ is injective by showing that its kernel is trivial. Let $A \in \ker(\Pi)$. Then, let $c, \hat{c} > \|A\|_{\mathrm{op}}$ and define $M = c^{-1}A + I$ and $\hat{M} = \hat{c}^{-1}A + I$, which are positive-definite by construction. Let $\rho$ and $\hat{\rho}$ be their corresponding Mahalanobis distances. Their subspace metrics on all $V_\lambda$'s coincide, since $A \in \ker(\Pi)$,

$$\Pi(M) = \Pi(c^{-1}A + I) = \Pi(I) = \Pi(\hat{c}^{-1}A + I) = \Pi(\hat{M}).$$

And so, by assumption $\rho = \hat{\rho}$. But as the matrix representation of a Mahalanobis distance is unique (Proposition D.1), this implies that $M = \hat{M}$, proving that $A = 0$. $\qquad\square$

# E  PROOFS AND ADDITIONAL RESULTS FOR SECTION 5

## E.1  PROOFS AND ADDITIONAL REMARKS FOR THEOREM 15

**Theorem 15.** *Let $\mathbb{R}^d$ have a Mahalanobis distance with matrix representation $M \in \mathrm{Sym}^+(\mathbb{R}^d)$. Let $\mathcal{X} \subset \mathbb{R}^d$ be subspace-clusterable over subspaces $V_\lambda$ indexed by $\lambda \in \Lambda$, where $|\Lambda| = n$. Let $\hat{M}$ be the estimator of $M$ and let $\hat{Q}_\lambda$ be the estimator of the subspace metric $Q_\lambda$ for each $\lambda$ learned from Algorithm 2. Suppose there exist $\gamma \leq \varepsilon$ such that $\|\mathbb{E}[\hat{Q}_\lambda] - Q_\lambda\|_{\mathrm{F}} \leq \gamma$ and $\|\hat{Q}_\lambda - Q_\lambda\|_{\mathrm{F}} \leq \varepsilon$ for each $\lambda$. Fix $p \in (0, 1]$. Then, there is a universal constant $c > 0$ such that with probability at least $1 - p$,*

$$\|\hat{M} - M\|_{\mathrm{F}} \leq c \cdot \frac{1}{\sigma_{\min}(\Pi)} \left( \gamma\sqrt{n} + \varepsilon d\sqrt{\log \frac{2d}{p}} \right),$$

*where $\sigma_{\min} > 0$ is the least singular value of $\Pi$.*

*Proof of Theorem 15.* Let $c = 2c_0$ where $c_0$ is a universal constant to be defined later. Recall from Eq. (5) that $\hat{M}$ minimizes $\|A - \hat{M}_{\mathrm{LS}}\|_{\mathrm{F}}$ over all $A \in \mathrm{Sym}^+(\mathbb{R}^d)$. Since $M$ is also contained in $\mathrm{Sym}^+(\mathbb{R}^d)$, we have:

$$\|\hat{M} - \hat{M}_{\mathrm{LS}}\|_{\mathrm{F}} \leq \|M - \hat{M}_{\mathrm{LS}}\|_{\mathrm{F}}.$$

By the triangle inequality,

$$\|\hat{M} - M\|_{\mathrm{F}} \leq \|\hat{M} - \hat{M}_{\mathrm{LS}}\|_{\mathrm{F}} + \|\hat{M}_{\mathrm{LS}} - M\|_{\mathrm{F}} \leq 2\|\hat{M}_{\mathrm{LS}} - M\|_{\mathrm{F}}.$$

Therefore, it suffices to show that, with probability $1 - \delta$,

$$\|\hat{M}_{\mathrm{LS}} - M\|_{\mathrm{F}} \leq c_0 \cdot \frac{1}{\sigma_{\min}(\Pi)} \left( \gamma\sqrt{m} + \varepsilon d\sqrt{\log \frac{2d}{\delta}} \right). \tag{13}$$

Before proving Eq. (13), we introduce some notation.

**Notation and facts**  For each subspace $V_\lambda$, we denote the recovery error by:

$$E_\lambda = \hat{Q}_\lambda - Q_\lambda = \Big( \underbrace{\mathbb{E}[\hat{Q}_\lambda] - Q_\lambda}_{H_\lambda \text{ (bias)}} \Big) + \Big( \underbrace{\hat{Q}_\lambda - \mathbb{E}[\hat{Q}_\lambda]}_{\xi_\lambda \text{ (noise)}} \Big),$$

which we decompose into a bias term $H_\lambda := \mathbb{E}[\hat{Q}_\lambda] - Q_\lambda$ and a noise term $\xi_\lambda := \hat{Q}_\lambda - \mathbb{E}[\hat{Q}_\lambda]$. By assumption,

$$\|H_\lambda\|_{\mathrm{F}} \leq \gamma \qquad \text{and} \qquad \mathbb{E}[\xi_\lambda] = 0, \quad \|\xi_\lambda\|_{\mathrm{F}} \leq \|E_\lambda\|_{\mathrm{F}} \leq \varepsilon.$$

Let $H := \bigoplus_{\lambda \in \Lambda} H_\lambda$, $\xi := \bigoplus_{\lambda \in \Lambda} \xi_\lambda$, and $E := H + \xi$. Thus, $E = \bigoplus_{\lambda \in \Lambda} \left( \hat{Q}_\lambda - Q_\lambda \right)$, by the above bias/noise decomposition. In addition, since $\|H_\lambda\|_{\mathrm{F}} \leq \gamma$, we have $\|H\| = \sqrt{\sum_{\lambda \in \Lambda} \|H_\lambda\|_{\mathrm{F}}^2} \leq \sqrt{m}\gamma$.

We now prove Eq. (13). Recall from Eq. (4) that $\hat{M}_{\mathrm{LS}}$ is the least-squares solution, so that:

$$
\begin{aligned}
\hat{M}_{\mathrm{LS}} - M &= \Pi^+(E) \\
&= \Pi^+(H + \xi),
\end{aligned}
\tag{14}
$$

where $\Pi^+ : \bigoplus_{\lambda \in \Lambda} \mathrm{Sym}(V_\lambda) \to \mathrm{Sym}(\mathbb{R}^d)$ denotes the Moore-Penrose inverse of $\Pi$ (see Definition B.2). It then follows from Eq. (14) and the triangle inequality that

$$
\left\| \hat{M}_{\mathrm{LS}} - M \right\|_{\mathrm{F}} \leq \left\| \Pi^+(H) \right\|_{\mathrm{F}} + \left\| \Pi^+(\xi) \right\|_{\mathrm{F}}.
\tag{15}
$$

By Proposition 13, the map $\Pi$ is injective since $\mathcal{X}$ is subspace-clusterable. Thus, $\sigma_{\min}(\Pi) > 0$, and:

$$
\left\| \Pi^+(H) \right\|_{\mathrm{F}} \leq \frac{1}{\sigma_{\min}(\Pi)} \|H\|_{\mathrm{F}} \leq \frac{1}{\sigma_{\min}(\Pi)} \gamma \sqrt{m}.
\tag{16}
$$

It then follows from Eq. (15) and Eq. (16) that, to prove Eq. (13), it suffices to show that, with probability at least $1 - \delta$,

$$
\left\| \Pi^+(\xi) \right\|_{\mathrm{F}} \leq c_0 \cdot \frac{1}{\sigma_{\min}(\Pi)} \left( \varepsilon d \sqrt{\log \frac{2d}{\delta}} \right).
\tag{17}
$$

By the universal property of the direct sum (see Proposition B.3), there exist $\Pi_\lambda^+ : \mathrm{Sym}(V_\lambda) \to \mathrm{Sym}(\mathbb{R}^d)$ for each $\lambda \in \Lambda$, such that

$$
\Pi^+(\xi) = \sum_{\lambda \in \Lambda} \Pi_\lambda^+(\xi_\lambda).
$$

Observe that

1. Each $\xi_\lambda$ is from subspace $V_\lambda$; and thus, $\xi_\lambda$'s and $\Pi_\lambda^+(\xi_\lambda)$'s across subspaces are independent,
2. $\mathbb{E}\left[ \Pi_\lambda^+(\xi_\lambda) \right] = \Pi_\lambda^+ \left( \mathbb{E}\left[ \xi_\lambda \right] \right) = 0$; and,
3. $\left\| \Pi_\lambda^+(\xi_\lambda) \right\|_{\mathrm{F}} \leq \left\| \Pi_\lambda^+ \right\|_{\mathrm{op}} \cdot \left\| \xi_\lambda \right\|_{\mathrm{F}} \leq \left\| \Pi_\lambda^+ \right\|_2 \cdot \varepsilon$,

where $\| \cdot \|_2$ denotes the 2-Schatten norm (see Definition B.4).

Corollary E.4 gives a Hoeffding-style concentration inequality for independent sub-Gaussian random matrices. Applied here, it states that there exists a universal constant $c_0$ such that, with probability $1 - \delta$,

$$
\begin{aligned}
\left\| \Pi^+(\xi) \right\|_{\mathrm{F}} &= \left\| \sum_{\lambda \in \Lambda} \Pi_\lambda^+(\xi_\lambda) \right\|_{\mathrm{F}} \\
&\overset{(i)}{\leq} c_0 \cdot \sqrt{\sum_{\lambda \in \Lambda} \left\| \Pi_\lambda^+ \right\|_2^2 \cdot \varepsilon^2 \log \frac{2d}{\delta}} \\
&\overset{(ii)}{=} c_0 \cdot \left\| \Pi^+ \right\|_2 \cdot \varepsilon \sqrt{\log \frac{2d}{\delta}} \\
&\overset{(iii)}{\leq} c_0 \cdot \frac{1}{\sigma_{\min}(\Pi)} \cdot \varepsilon d \sqrt{\log \frac{2d}{\delta}},
\end{aligned}
\tag{18}
$$

where (i) applies the third observation from above, (ii) applies Proposition B.5 about 2-Schatten norms, and (iii) uses the following facts:

- $\|\Pi^+\|_2 \leq \sigma_{\max}(\Pi^+) \cdot \sqrt{\mathrm{rank}\,(\Pi^+)}$, (see Definition B.4),
- $\sigma_{\max}(\Pi^+) = \dfrac{1}{\sigma_{\min}(\Pi)}$,
- $\mathrm{rank}(\Pi^+) \leq \frac{d(d+1)}{2} \leq d^2$.

$\square$

## E.2 PROOFS AND ADDITIONAL REMARKS FOR PROPOSITION 19

**Proposition 19** (Theorem 4.1, Canal et al. [2022]). *Suppose that $\mathbb{R}^r$ has a Mahalanobis distance with representation $Q \in \mathrm{Sym}^+(\mathbb{R}^r)$ where $\|Q\|_{\mathrm{F}} \leq \zeta_M$. Let each user $k \in [K]$ have pseudo-ideal point $v_k \in \mathbb{R}^r$ where $v_k \leq \zeta_v$. Let $\mathcal{P}_m$ be a distribution over designs of size $m$ over $\mathbb{R}^r$ (Definition 1). For each user, let $D_k \sim \mathcal{P}_m$ be an i.i.d. random design, and let $\mathcal{D}_k = \{(x_{i_0}, x_{i_1}, y_{i;k})\}_{i \in [m]}$ be the user's responses under Assumption 17. Fix $p \in (0, 1]$. Given loss function $\ell(z) = -\log f(z)$, Algorithm 5 returns $\hat{Q} \in \mathrm{Sym}^+(\mathbb{R}^r)$, where with probability at least $1 - p$,*

$$\|\hat{Q} - Q\|_{\mathrm{F}}^2 \leq \frac{16L}{c_f^2 \cdot \sigma_{\min}^2(\mathcal{P}_m)} \sqrt{\frac{(\zeta_M^2 + K\zeta_v^2) \log \frac{4}{p}}{mK}}.$$

*Proof.* The objective over which the parameters $(A, w_1, \dots, w_K)$ is optimized in Eq. (7) of Algorithm 5 can be written as:

$$\hat{R}(A, w_1, \dots, w_K) = \sum_{k \in [K]} \sum_{i \in [m]} -\log f(y_{i;k} \cdot D_{i;k}(A, w_k)).$$

Let $(\hat{Q}, \hat{v}_1, \dots, \hat{v}_K)$ be the solution recovered in this step of Algorithm 5. The excess risk of these parameters is defined to be how much worse in expectation the parameters are at explaining observed data compared to the true parameters $(Q, v_1, \dots, v_K)$ that generated the data. The excess risk leads to a bound on $\|\hat{Q} - Q\|_{\mathrm{F}}^2$,

$$\mathbb{E}\left[\hat{R}(\hat{Q}, \hat{v}_1, \dots, \hat{v}_K)\right] - \mathbb{E}\left[\hat{R}(Q, v_1, \dots, v_K)\right]$$

$$\stackrel{(a)}{=} \sum_{k \in [K]} \mathbb{E}_{D_k \sim \mathcal{P}_m} \left[\sum_{i \in [m]} \mathrm{KL}\left(f(D_{i;k}(Q, v_k)) \,\Big\|\, f(D_{i;k}(\hat{Q}, \hat{v}_k))\right)\right]$$

$$\stackrel{(b)}{\geq} 2c_f^2 \sum_{k \in [K]} \mathbb{E}_{D_k \sim \mathcal{P}_m} \left\|D_k(\hat{Q} - Q, \hat{v}_k - v_k)\right\|^2$$

$$\stackrel{(c)}{\geq} 2c_f^2 \sum_{k \in [K]} m \cdot \sigma_{\min}^2(\mathcal{P}_m) \cdot \left(\|\hat{Q} - Q\|_{\mathrm{F}}^2 + \|\hat{v}_k - v_k\|^2\right)$$

$$\geq 2mKc_f^2 \cdot \sigma_{\min}^2(\mathcal{P}_m) \cdot \|\hat{Q} - Q\|_{\mathrm{F}}^2, \tag{19}$$

where each inequality is justified below. We just need to show that the excess risk of $\hat{Q}$ returned by the algorithm has small excess risk. Lemma E.1 approaches this via a standard generalization argument, showing that with probability at least $1 - \delta$,

$$\mathbb{E}\left[\hat{R}(\hat{Q}, \hat{v}_1, \dots, \hat{v}_K)\right] - \mathbb{E}\left[\hat{R}(Q, v_1, \dots, v_K)\right]$$

$$\leq \underbrace{\hat{R}(\hat{Q}, \hat{v}_1, \dots, \hat{v}_K) - \hat{R}(Q, v_1, \dots, v_K)}_{\leq 0} + 32L\sqrt{mK(\zeta_M^2 + K\zeta_v^2) \log \frac{4}{\delta}}, \tag{20}$$

where the indicated difference is less than zero because $(\hat{Q}, \hat{v}_1, \dots, \hat{v}_k)$ is the minimizer of $\hat{R}$. The result is obtained by combining Eqs. (19) and (20). To finish the prove, we justify the above inequalities:

(a) Recall that $\Pr[Y_{i;k} = +1] = f(D_{i;k}(Q, v_k))$. Because $f(z) = 1 - f(-z)$, we also have that:

$$\Pr[Y_{i;k} = -1] = 1 - f(D_{i;k}(Q, v_k)) = f(-D_{i;k}(Q, v_k)).$$

Therefore, $\Pr[Y_{i;k} = y] = f(y \cdot D_{i;k}(Q, v_k))$. It follows that the excess risk is equal to:

$$\mathbb{E}\left[\hat{R}(\hat{Q}, \hat{v}_1, \dots, \hat{v}_K)\right] - \mathbb{E}\left[\hat{R}(Q, v_1, \dots, v_K)\right]$$

$$= \sum_{k \in [K]} \mathbb{E}_{D_k, Y} \left[\sum_{i \in [m]} -\log\left(\frac{f(Y_{i;k} \cdot D_{i;k}(Q, v_k))}{f(Y_{i;k} \cdot D_{i;k}(\hat{Q}, \hat{v}_k))}\right)\right]$$

$$= \sum_{k \in [K]} \mathbb{E}_{D_k} \left[\sum_{i \in [m]} \sum_{y \in \{-1, +1\}} -f(y \cdot D_{i;k}(Q, v_k)) \log\left(\frac{f(y \cdot D_{i;k}(Q, v_k))}{f(y \cdot D_{i;k}(\hat{Q}, \hat{v}_k))}\right)\right],$$

where we obtain the equality (a) by applying the definition $\mathrm{KL}(p\|q)$,

$$\mathrm{KL}(p\|q) = p\log\frac{p}{q} + (1-p)\log\frac{1-p}{1-q}.$$

(b) The following is the same argument used in [Canal et al., 2022, Proposition E.3].

$$
\begin{aligned}
\sum_{i\in[m]} \mathrm{KL}\left(f\big(D_{i;k}(Q,v_k)\big)\,\Big\|\,f\big(D_{i;k}(\hat{Q},\hat{v}_k)\big)\right) &\geq 2\sum_{i\in[m]}\left(f\big(D_{i;k}(Q,v_k)\big) - f\big(D_{i;k}(\hat{Q},\hat{v}_k)\big)\right)^2 \\
&\geq 2c_f^2\sum_{i\in[m]}\left(D_{i;k}(Q,v_k) - D_{i;k}(\hat{Q},\hat{v}_k)\right)^2 \\
&= 2c_f^2\sum_{i\in[m]}\left(D_{i;k}(\hat{Q}-Q,\hat{v}_k-v_k)\right)^2 \\
&= 2c_f^2\left\|D_k(\hat{Q}-Q,\hat{v}_k-v_k)\right\|^2,
\end{aligned}
$$

where the first inequality comes from $\mathrm{KL}(p\|q)\geq 2(p-q)^2$, see [Mason et al., 2017, Lemma 5.2], the second uses the monotonicity of $f$ and the lower bound of $f'$, the third applies linearity of $D_{i;k}$, and the fourth just rewrites the sum in terms of the squared $\ell_2$-norm over $\mathbb{R}^m$.

(c) Recall that $\sigma^2(\mathcal{P}_m) = \frac{1}{m}\cdot\sigma_{\min}(\mathbb{E}[D^*D])$ when $D\sim\mathcal{P}_m$. Let $X = (\hat{Q}-Q)\oplus(\hat{v}_k-v_k)$ for short. Then,

$$
\begin{aligned}
\mathbb{E}\left\|D_k(\hat{Q}-Q,\hat{v}_k-v_k)\right\|^2 &= \mathbb{E}\left\langle D_kX, D_kX\right\rangle \\
&= X^\top\,\mathbb{E}\left[D_k^*D_k\right]X \\
&\geq \sigma_{\min}\left(\mathbb{E}\left[D_k^*D_k\right]\right)\cdot\|X\|^2 \\
&= m\cdot\sigma_{\min}^2(\mathcal{P}_m)\cdot\left(\|\hat{Q}-Q\|_{\mathrm{F}}^2 + \|\hat{v}_k-v_k\|^2\right),
\end{aligned}
$$

where the inequality applies the variational characterization of the minimum singular value.

$\square$

**Lemma E.1.** *Let $\delta\in(0,1)$. Given the assumptions of Proposition 19, Eq. (20) holds with probability at least $1-\delta$.*

*Proof.* For short, let $\Theta\subset\mathrm{Sym}^+(\mathbb{R}^r)\oplus\mathbb{R}^{r\times K}$ denote the set of parameters $\theta\equiv(A,w_1,\ldots,w_K)$ such that $A\in\mathrm{Sym}^+(\mathbb{R}^r)$ with $\|A\|_{\mathrm{F}}\leq\zeta_M$ and $w_k\in\mathbb{R}^r$ with $\|w_k\|\leq\zeta_v$. We claim that with probability at least $1-\delta$, we have uniform convergence:

$$\sup_{\theta\in\Theta}\left|\hat{R}(\theta) - \mathbb{E}\left[\hat{R}(\theta)\right]\right| \leq 16L\sqrt{mK(\zeta_M^2 + K\zeta_v^2)\log\frac{4}{\delta}}. \tag{21}$$

Before proving this, notice that this implies Eq. 20. In particular, let $\hat{\theta}$ correspond to the parameters $(\hat{Q},\hat{v}_1,\ldots,\hat{v}_K)$ and let $\theta$ correspond to $(Q,v_1,\ldots,v_K)$. Then we have that with probability at least $1-\delta$, both $\hat{R}(\hat{\theta})$ and $\hat{R}(\theta)$ are close to their expected values, each contributing at most the right-hand side of Eq. (21):

$$\mathbb{E}\left[\hat{R}(\hat{\theta})\right] - \mathbb{E}\left[\hat{R}(\theta)\right] \leq \hat{R}(\hat{\theta}) - \hat{R}(\theta) + 32L\sqrt{mK(\zeta_M^2 + K\zeta_v^2)\log\frac{4}{\delta}}.$$

In the remainder of the proof, we show Eq. (21). For any $\theta\in\Theta$, consider the empirical risk $\hat{R}(\theta)$. We claim that the risk contribution by the $i$th comparison by the $k$th user is a bounded random variable,

$$\left|-\log\left(f\big(Y_{i;k}\cdot D_{i;k}(A,w_k)\big)\right) + \log\frac{1}{2}\right| \overset{(a)}{\leq} 2L(\zeta_M + \zeta_v).$$

Let us verify this claim later. For now, the bounded difference inequality (reproduced below as Lemma E.5) implies that with probability at least $1 - \delta$,

$$\sup_{\theta \in \Theta} \left| \hat{R}(\theta) - \mathbb{E}\left[\hat{R}(\theta)\right] \right| \le \mathbb{E}\left[ \sup_{\theta \in \Theta} \left| \hat{R}(\theta) - \mathbb{E}\left[\hat{R}(\theta)\right] \right| \right] + 4L(\zeta_M + \zeta_v)\sqrt{2mK \log \frac{2}{\delta}}. \tag{22}$$

To bound the expectation term, let us combine each user's random design matrix $D_k$ into a single $(m, K)$-experimental design matrix $D : \mathrm{Sym}(\mathbb{R}^r) \oplus \mathbb{R}^{r \times K} \to \mathbb{R}^{m \times K}$, so that it is the following linear map:

$$D(A, w_1, \ldots, w_K)_{i;k} = D_{i;k}(A, w_k).$$

Let $D^* : \mathbb{R}^{m \times K} \to \mathrm{Sym}(\mathbb{R}^r) \oplus \mathbb{R}^{m \times K}$ be its adjoint. Let $\epsilon \in_R \{-1, +1\}^{m \times K}$ be an array of independent Rademacher random variables, so that $\epsilon_{i;k}$ is equal to $-1$ or $+1$ uniformly at random. Then:

$$\mathbb{E}\left[ \sup_{\theta \in \Theta} \left| \hat{R}(\theta) - \mathbb{E}\left[\hat{R}(\theta)\right] \right| \right] \overset{(b)}{\le} 2\,\mathbb{E}\left[ \sup_{A, w_1, \ldots, w_K} \left| \sum_{k \in [K]} \sum_{i \in [m]} \epsilon_{i;k} \left( -\log f\left(Y_{i;k} \cdot D_{i;k}(A, w_k)\right)\right) \right| \right]$$

$$\overset{(c)}{\le} 2L \cdot \mathbb{E}\left[ \sup_{A, w_1, \ldots, w_K} \left| \sum_{k \in [K]} \sum_{i \in [m]} \epsilon_{i;k} \left(Y_{i;k} \cdot D_{i;k}(A, w_k)\right) \right| \right]$$

$$\overset{(d)}{=} 2L \cdot \mathbb{E}\left[ \sup_{\theta \in \Theta} \left| \langle \epsilon, D(\theta) \rangle \right| \right]$$

$$\overset{(e)}{\le} 2L \cdot \mathbb{E} \left\| D^* \epsilon \right\| \cdot \sup_{\theta \in \Theta} \|\theta\|$$

$$\overset{(f)}{\le} 4L\sqrt{2mK(\zeta_M^2 + K\zeta_v^2)}, \tag{23}$$

where we justify each step below. We obtain Eq. (21) by combining Eqs. (22) and (23),

$$\sup_{\theta \in \Theta} \left| \hat{R}(\theta) - \mathbb{E}\left[\hat{R}(\theta)\right] \right| \le 4L\sqrt{2mK(\zeta_M^2 + K\zeta_v^2)} + 4L(\zeta_M + \zeta_v)\sqrt{2mK \log \frac{2}{\delta}}$$

$$\overset{(i)}{\le} 4L\sqrt{2\left( 2mK(\zeta_M^2 + K\zeta_v^2) + 2mK(\zeta_m + \zeta_v)^2 \log \frac{2}{\delta} \right)}$$

$$\overset{(ii)}{\le} 8L\sqrt{mK \cdot (\zeta_m^2 + K\zeta_v^2) \cdot \left(1 + 3\log \frac{2}{\delta}\right)}$$

$$\overset{(iii)}{\le} 16L\sqrt{mK \cdot (\zeta_M^2 + K\zeta_v^2) \log \frac{4}{\delta}},$$

where (i) applies a variant of the AM-GM inequality $\sqrt{a} + \sqrt{b} \le \sqrt{2(a+b)}$, (ii) uses the following upper bound $(\zeta_M + \zeta_v)^2 \le 3(\zeta_M^2 + K\zeta_v^2)$, which holds whenever $\zeta_M, \zeta_v \ge 0$ and $K \ge 1$, and (iii) uses $1 < 3\log 2$ and $8\sqrt{3} < 16$. Finally, we prove the remaining inequalities:

(a) Because we have assumed that items lie in the unit ball and that the parameters satisfy $\|A\|_F \le \zeta_M$ and $\|w_i\| \le \zeta_v$, the unquantized measurements are bounded:

$$\left| D_{i;k}(A, v_k) \right| \le \sup_{x, x' \in B(0,1)} \left| \langle xx^\top - x'x'^\top, A \rangle + \langle x - x', v_k \rangle \right| \le 2\|A\|_F + 2\|v_k\| \le 2(\zeta_M + \zeta_v),$$

where we have used triangle inequality for $\left\| xx^\top - x'x'^\top \right\|_F \le 2$ and $\|x - x'\| \le 2$. Because $-\log f(\cdot)$ is $L$-Lipschitz on this domain, whenever $|z| \le 2(\zeta_M + \zeta_v)$, we have:

$$\left| -\log f(z) + \log \frac{1}{2} \right| = \left| -\log f(z) + \log f(0) \right| \le L|z|.$$

(b) This inequality follows from a standard symmetrization argument. Let $\mathcal{H}$ be a set of $N$-tuples of functions, where $h \equiv (h_1, \ldots, h_N)$. Given a set of i.i.d. random variables $Z_1, \ldots, Z_N, Z_1', \ldots, Z_N'$ and a set of Rademacher random variables $\epsilon_1, \ldots, \epsilon_N \in \{-1, +1\}$, we have:

$$
\mathbb{E}\left[\sup_{h \in \mathcal{H}} \left| \sum_{i=1}^{N} h_i(Z_i) - \mathbb{E}\left[\sum_{i=1}^{N} h_i(Z_i)\right] \right| \right] = \mathbb{E}\left[\sup_{h \in \mathcal{H}} \left| \sum_{i=1}^{N} \epsilon_i h_i(Z_i) - \sum_{i=1}^{N} \epsilon_i h_i(Z_i') \right| \right]
$$

$$
\leq \mathbb{E}\left[\sup_{h \in \mathcal{H}} \left| \sum_{i=1}^{N} \epsilon_i h(Z_i) \right| \right] + \mathbb{E}\left[\sup_{h \in \mathcal{H}} \left| \sum_{i=1}^{N} \epsilon_i h_i(Z_i') \right| \right]
$$

$$
= 2\mathbb{E}\left[\sup_{h \in \mathcal{H}} \left| \sum_{i=1}^{N} \epsilon_i h_i(Z_i) \right| \right].
$$

In our setting, we have an index set $(i, k) \in [m] \times [K]$ and $h_{i;k} : Z \mapsto -\log f\big(Z \cdot D_{i;k}(A, w_k)\big)$.

(c) We use the fact that the function $-\log f(z)$ is $L$-Lipschitz over the domain $|z| \leq 2(\zeta_M + \zeta_v)$. We can move the Lipschitz constant out of the expectation by applying [Zhang, 2023, Theorem 6.28], reproduced below.

(d) This step first makes use of the fact that the random variables $\epsilon_{i;k} Y_{i;k} \stackrel{d}{=} \epsilon_{i;k}$ are equal in distribution. Then, it consolidates everything using the trace inner product on $\mathbb{R}^{m \times K}$.

(e) This step uses the property of the adjoint $\langle \epsilon, D(\theta) \rangle = \langle D^*(\epsilon), \theta \rangle \leq \|D^*(\epsilon)\| \cdot \|\theta\|$. The first inner product is over $\mathbb{R}^{m \times K}$, the second inner product and norm are over $\mathrm{Sym}(\mathbb{R}^r) \otimes \mathbb{R}^{r \times K}$.

(f) We apply the bound on the parameters $\sup_{\theta \in \Theta} \|\theta\| \leq \sqrt{\zeta_M^2 + K\zeta_v^2}$ along with the following:

$$
\mathbb{E}\left\| D^* \epsilon \right\| \stackrel{(i)}{\leq} \sqrt{\mathbb{E}\left\langle DD^*, \epsilon\epsilon^\top \right\rangle}
$$

$$
\stackrel{(ii)}{=} \sqrt{\left\langle \mathbb{E}[DD^*], \mathbb{E}[\epsilon\epsilon^\top] \right\rangle}
$$

$$
\stackrel{(iii)}{=} \sqrt{\sum_{i,k} \left\| \Delta_{i;k} \oplus \delta_{i;k} \right\|^2} \stackrel{(iv)}{\leq} 2\sqrt{2mK}.
$$

The (i) uses Jensen's inequality, (ii) uses the independence of the randomness over the design matrices and the Rademacher random variables, (iii) uses the fact that $\mathbb{E}[\epsilon\epsilon^\top]$ is the identity on $\mathbb{R}^{m \times K}$, and (iv) uses the fact that items are contained in the unit Euclidean ball, so that:

$$
\|\Delta_{i;k} \oplus \delta_{i;k}\|^2 = \|\Delta_{i;k}\|^2 + \|\delta_{i;k}\|^2 \leq 2^2 + 2^2.
$$

$\square$

**Remark E.2.** *To show that there exists $\mathcal{P}_m$ such that $\sigma_{\min}^2(\mathcal{P}_m) = \Omega(1)$, assume the space $\mathbb{R}^r$ is quadratically spanned by $\mathcal{X}$. In particular, there exists a collection of items $(x_{i_0}, x_{i_1})_{i=1}^n$ such that its design matrix $D$ is full rank. Define $X_i \in \mathrm{Sym}(\mathbb{R}^r) \oplus \mathbb{R}^r$ for $i = 1, \ldots, n$ by $X_i = \Delta_i \oplus \delta_i$. Then, $D^* D$ corresponds to:*

$$
D^* D = \sum_{i=1}^{n} X_i X_i^\top,
$$

*where $\sigma_{\min}(D^* D) > 0$. Let $\mathcal{P}_m$ be constructed by drawing $m$ pairs uniformly at random. Let $D_m$ be the random design*

*matrix. Let $I_j \sim \text{Unif}([n])$ for $j = 1, \ldots, m$ be the index of the $j$th random pair, so that:*

$$\mathbb{E}[D_m^* D_m] = \mathbb{E}\left[\sum_{i=1}^m X_{I_j} X_{I_j}^\top\right]$$

$$= \sum_{j=1}^m \frac{1}{n} \sum_{i=1}^n X_i X_i^\top$$

$$= \frac{m}{n} D^* D.$$

*It follows that for this choice of random design, we have $\sigma_{\min}^2(\mathcal{P}_m) = \sigma_{\min}(D^* D)$, which is a constant.*

### E.3 AUXILIARY LEMMAS

**Lemma E.3** (Hoeffding-style inequality for independent bounded random vectors, [Jin et al., 2019], Corollary 7)**.** *There exists a universal constant $c$ such that for any random vectors $X_1, X_2, \ldots, X_m \in \mathbb{R}^d$ that are independent and satisfy $\mathbb{E}[X_i] = 0$ and $\|X_i\|_2 \leq \kappa_i$ for $i \in [m]$, we have, for any $\delta \in (0, 1]$, with probability at least $1 - \delta$,*

$$\left\|\sum_{i=1}^m X_i\right\|_2 \leq c \cdot \sqrt{\sum_{i=1}^m \kappa_i^2 \log \frac{2d}{\delta}}.$$

**Corollary E.4** (Matrix version, [Jin et al., 2019], Corollary 7)**.** *There exists a universal constant $c$ such that for any random matrices $X_1, X_2, \ldots, X_m \in \mathbb{R}^{d \times d}$ that are independent and satisfy $\mathbb{E}[X_i] = 0$ and $\|X_i\|_{\mathrm{F}} \leq \kappa_i$ for $i \in [m]$, we have, for any $\delta \in (0, 1]$, with probability at least $1 - \delta$,*

$$\left\|\sum_{i=1}^m X_i\right\|_{\mathrm{F}} \leq c \cdot \sqrt{\sum_{i=1}^m \kappa_i^2 \log \frac{2d}{\delta}}.$$

*Proof.* Since $\log\left(\frac{2d^2}{\delta}\right) \leq 2 \log\left(\frac{2d}{\delta}\right)$ for $\delta \leq 1$, the corollary follows directly from Lemma E.3. $\qquad\square$

**Lemma E.5** (Bounded difference inequality)**.** *Let $f : \mathcal{X}^N \to \mathbb{R}$ satisfy the bounded difference property,*

$$\sup_{x_1, \ldots, x_N, x_i'} \left|f(x_1, \ldots, x_N) - f(x_1, \ldots, x_i', \ldots, x_N)\right| \leq C, \qquad \forall i \in [N].$$

*Let $X_1, \ldots, X_N$ be i.i.d. random variables. Then, with probability at least $1 - \delta$,*

$$\left|f(X_1, \ldots, X_N) - \mathbb{E}[f(X_1, \ldots, X_N)]\right| \leq C\sqrt{2N \log \frac{2}{\delta}}.$$

This theorem is also known as McDiarmid's inequality; as reference, see for example [Zhang, 2023, Theorem 6.16].

**Lemma E.6** (Theorem 6.28, [Zhang, 2023])**.** *Let $h$ be an $L$-Lipschitz function $h : \mathbb{R} \to \mathbb{R}$. Let $\mathcal{F}$ be a function class with functions $f : \mathcal{Z} \to \mathbb{R}$. Let $z_1, \ldots, z_N \in \mathcal{Z}$ and let $\epsilon_1, \ldots, \epsilon_N$ be independent Rademacher random variables. Then:*

$$\mathbb{E}\left[\sup_{f \in \mathcal{F}} \left|\sum_{i=1}^N \epsilon_i h\big(f(z_i)\big)\right|\right] \leq L \cdot \mathbb{E}\left[\sup_{f \in \mathcal{F}} \left|\sum_{i=1}^N \epsilon_i f(z_i)\right|\right].$$

# F DETAILS AND ADDITIONAL RESULTS FOR SECTION 6

Our experimental setup and implementation are inspired by and adapted from [Canal et al., 2022]. In Section F.1, we provide further details to our experimental setup. In Section F.2, we present additional experimental results.

## F.1 EXPERIMENTAL DETAILS

Each simulation run is defined by several parameters: the ambient dimension $d$, the number of subspaces $n$, the dimension of each subspace $r$, the number of users per subspace $K$, and the number of preference comparisons per user $m$.

**Data generation** In each simulation run, we generate a symmetric positive definite matrix $M$ from the Wishart distribution $W(d, I_d)$ and normalize it so that $\|M\|_{\mathrm{F}} = d$, as in [Canal et al., 2022, Section F.3]. We generate $n$ uniform-at-random $r$-dimensional subspaces [Stewart, 1980]: for each subspace, we draw $r$ independent random vectors from $\mathcal{N}(0, \frac{1}{d}I_d)$ and use QR decomposition to find an orthonormal basis. For each subspace $V$ equipped with orthonormal basis $B$, we generate $K$ user ideal points from $\mathcal{N}(0, \frac{1}{d}I_d)$. For each user, we generate $2m$ items ($m$ pairs), where each item is a fresh draw from $\mathcal{N}(0, \frac{1}{r}BB^\top)$.

For Experiment 3, given $V$ and $B$ generated in this way, we generate $2mK$ items that approximately lie on a subspace $V$ by sampling from $\mathcal{N}(0, \frac{1}{r}BB^\top + \frac{\sigma^2}{d-r}B_\perp B_\perp^\top)$, where $B_\perp$ is an orthonormal basis of $V^\perp$, the orthogonal complement of $V$. We generate user responses as before (see Section 6; namely, we use the sigmoid link function with varying choices of noise levels, $\beta = 1, 4, \infty$).

To learn the metric using these approximately subspace-clusterable items, Algorithm 2 needs to be modified since it expects that the items lie exactly on a union of subspaces. We do so by constructing new representations for the items, where for each set of $2mK$ items $X \in \mathbb{R}^{d \times 2mK}$ that approximately lie on $V$, we use singular value decomposition to construct a rank-$r$ approximation $\hat{X}$ of the items, minimizing:

$$\min_{\mathrm{rank}(\hat{X}) \leq r} \|\hat{X} - X\|_F.$$

See the Eckart-Young-Mirsky theorem [Golub et al., 1987, for reference]. Algorithm 2 can then be run directly on the low-rank representation $\hat{X}$ (this procedure does not affect how user responses are generated).

**Algorithm implementation** We provide additional details on the implementation of Algorithm 2. In Stage 1 (learning subspace metrics), we use Algorithm 5 and set constraints based on oracle knowledge of optimal hyperparameters $\zeta_M$ and $\zeta_v$ (also called the best-case hyperparameter setting in [Canal et al., 2022]). We use $\ell(z; \beta) = \log(1 + \exp(-\beta z))$ as the loss function, where $\beta$ is assumed known and given by the logistic link function above. We use the Splitting Conic Solver (SCS) in CVXPy with hyperparameters `eps = 1e4` and `max_iters = 1e5` to solve the convex optimization problem.

In Stage 2 of our practical implementation (reconstruction from subspace metrics), we note that least squares can be sensitive to outliers, and therefore we use the Huber loss for robust regression [Huber, 1964]. In particular, we use the HuberRegressor from scikit-learn [Pedregosa et al., 2011] with default hyperparameters, except for setting `max_iters = 1e4`. To reconstruct a full metric, we use subspace metrics learned in Stage 1. We note that we do not include a subspace (and the learned subspace metric) into our reconstruction step if CVXPy/SCS does not solve the corresponding optimization problem in Stage 1 successfully, that is, `prob.status != OPTIMAL`. Nevertheless, given $n$ subspaces, if CVXPy/SCS does not successfully solve any of them, we use the $n$-th subspace alone for reconstruction.

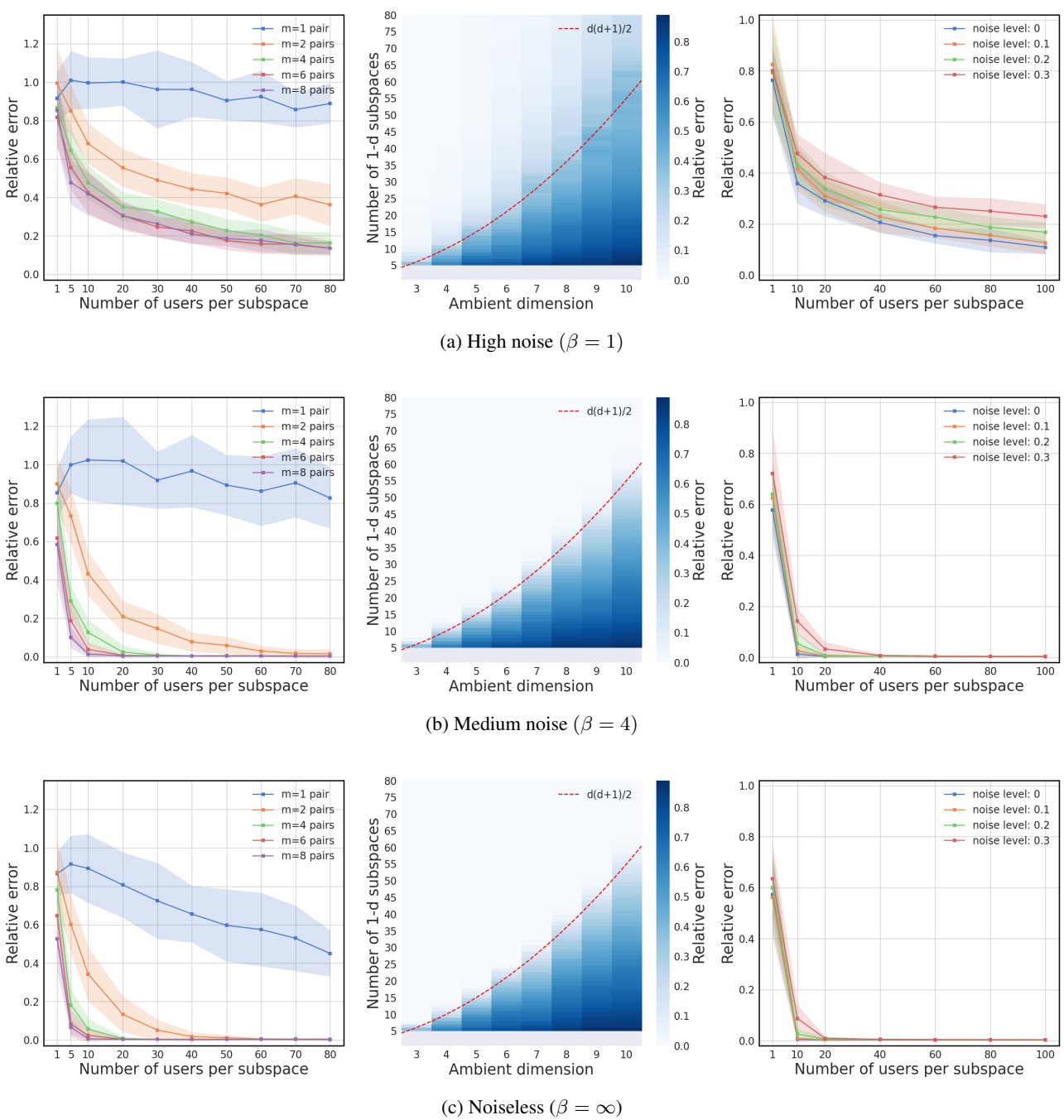

(a) High noise $(\beta = 1)$

(b) Medium noise $(\beta = 4)$

(c) Noiseless $(\beta = \infty)$

Figure F.1: shows the results obtained using the same data in the three experiments (Section 6), wherein the learner now uses the hinge loss to recover subspace metrics (Algorithm 5). Note that the $y$-axis scales in the plots for Experiment 3 have been slightly adjusted to enhance clarity.

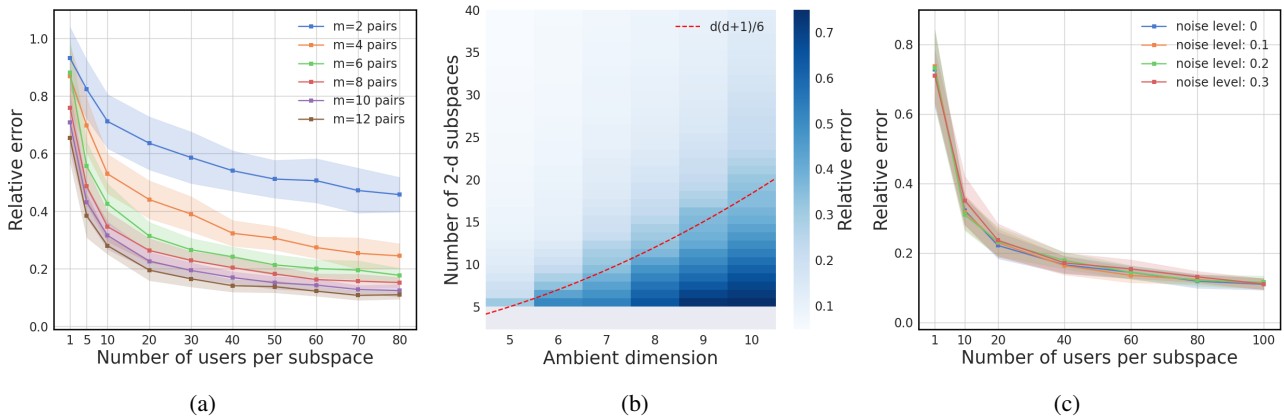

(a)                (b)                (c)

Figure F.2: (a) shows the average relative errors over items that lie in a union of 40 2-dimensional subspaces. (b) shows the average relative errors for reconstructing $\hat{M}$ from increasing numbers of 2-dimensional subspaces; for each subspace, 80 users each provides 10 preference comparisons. The dotted red curve illustrates the counting argument in Remark 14; here, each 2-dimensional subspace can contribute at most 3 degrees of freedom. (c) shows the average relative errors for varying subspace noise levels; here, items lie approximately in a union of 40 2-dimensional subspaces and each user provides 10 preference comparisons.

## F.2   ADDITIONAL EXPERIMENTAL RESULTS

We further study whether our two-stage approach requires exact knowledge of the probabilistic model under which user binary responses are generated. To this end, we repeated the three experiments in Section 6 where user responses are sampled according to the logistic sigmoid link function with varying response noise levels, $\beta = 1$, $\beta = 4$, and $\beta = \infty$ (noiseless). Given the same data used for the experiments discussed in Section 6, we now set the learner to use the hinge loss,

$$\ell(z) = \max\left(0, 1 - z\right),$$

instead of the negative log loss, to learn subspace metrics in Stage 1 of Algorithm 5. Figure F.1 shows the performance of the learner. When compared with the results in Figures 3 and 4, the learner still recovers the full metric reasonably well. This further validates the effectiveness of our divide-and-conquer approach.

We also ran the three experiments in Section 6 for subspace dimension $r = 2$, with slightly different parameters. The response noise level was set to $\beta = 1$ and was known to the learner, and each experiment was run 30 times. Figure F.2a compares the average relative errors for varying $K$ and $m$, where items lie in a union of 40 subspaces. Figure F.2b shows the average errors given increasing numbers of subspaces, where $K = 80$ and $m = 10$. Note that by the dimension-counting argument in Remark 14, each 2-dimensional subspace contributes at most $\frac{2(2+1)}{2} = 3$ degrees of freedom, and therefore a minimum of $\left\lceil \frac{d(d+1)}{6} \right\rceil$ subspaces are needed. Figure F.2b shows the average recovery errors for varying subspace noise levels, $\sigma \in \{0, 0.1, 0.2, 0.3\}$, and varying $K$, where items lie in a union of 40 subspaces and we set $m = 10$.

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
