# OpenReview forum: "Metric Learning from Limited Pairwise Preference Comparisons"
_auai.org/UAI/2024/Conference — UAI 2024 poster_

### Official Review · Reviewer_of7f · 2024-03-07

**Q2-1 Originality-Novelty:** 3
**Q2-2 Correctness-Technical Quality:** 3
**Q2-5 Clarity Of Writing:** 3

**Q1 Summary And Contributions:**

The authors try to learn a metric distance (Mahalanobis distance) from limited pairwise preference comparisons. The idea is to determine a Mahalanobis distance (or more accurately, the matrix M) through pairwise preference comparisons given by different users, supposing the different users use the same distance to measure the distance between their ideal point and the different items. They cannot give unlimited comparisons, as it is unrealistic (heavy cognitive effort) and can be antagonistic with respect to privacy.

In the literature, works suggest it is possible to recover both the metric and the ideal items through O(d) pairwise comparisons per user. However, the authors note that realistically we can have only o(d) pairwise comparisons, and they prove mathematically that, in the general case, o(d) comparisons are not enough to get any of the information, even with an infinite number of users.

Additionally, the authors proposed a method to bypass these limits in many cases by using low-rank subspaces. More simply said, they use a divide-and-conquer approach by comparing items in low-dimensional spaces and joining the spaces in order to recover the original space. After justifying their approach theoretically in both precise and noisy (preferences) cases, they show some synthetic experiments where their approach is indeed effective to estimate the distance, even in the noisy case.

**Q2-3 Extent To Which Claims Are Supported By Evidence:**

4: Excellent: all claims are supported by very convincing evidence (in the form of comprehensive experimental evaluation, rigorous mathematical proofs, detailed (pseudo-)code, precise references, well-motivated and realistic assumptions) and the authors deliver what they promise.

**Q2-4 Reproducibility:**

3: Good: key resources (e.g. proofs, code, data) are available and key details (e.g. proofs, experimental setup) are sufficiently well-described for competent researchers to confidently reproduce the main results.

**Q3 Main Strengths:**

The claims in this paper are clearly very supported by evidence. The authors provided an impressive number of rigorous mathematical proofs, pseudocode, precise references, etc., most of them being in their supplementary material due to limited space. The evidences are each time clearly linked with the main paper, so the readers know where to find the proofs. Despite the big supplementary material, I still find the main paper to be self-sufficient. Each time they proposed something, they made sure to provide a rigorous theoretical justification. In addition, references to other works are pertinent, as they are used to justifying why they made some assumptions, presenting other methods from the literature, etc. Additionally, the experiments also provide good evidence to their claims, even if they can look more limited compared to the theoretical proofs.

Sadly, I cannot assert if the paper is excellent regarding technical quality, as I am unable to check all the details - I do not have the sufficient mathematical background. From what I was able to check, the paper seems to be very technically sound. Regardless of my limitations, I would like to point that the authors tried to give as much background as possible, which helped me a lot to understand even their most technical points.

**Q4 Main Weakness:**

There are, to my knowledge, no main weaknesses. There are minor points I'll detail in Q5, but nothing really major.

**Q5 Detailed Comments To The Authors:**

Minor:
- I would perhaps highlight even more in the abstract the fact that you are working with uncertainty, especially with noisy preferences in at least one setting.
- Could you explain somewhere the difference between o and O (and Theta) somewhere in the supplementary or in the paper (or with simpler equivalents)? Having a computer science background, I am quite familiar with these asymptotic notations and I think I understood what they represented (with O we can have d comparisons, with o never exactly d), but that could be interesting for people without detailed knowledge on computation complexity.
- In the related work, you could also add that comparing triplets (even without a ranking of them) is cognitively harder for the user, explaining why most works even in different preference learning fields prefer indeed to compare only two alternatives.
- In Page 3, you talk about K without introducing it properly (which you do way later, in the experiments if I recall).
- A link to your code for the experiments would be perfect, even if the reproducibility is already good enough, given the amount of details available about the protocol.
- Figures 2 a) and c) are not colorblind friendly. I would change the line shapes depending on m and the noise level, so even in black and white we can correctly analyze the plots. Indeed, in the figure a) m=1 is the line at the top, while in figure c) noise level 0 is at the bottom, and without colors we would make erroneous observations.
- Figure 2 b), does the white means 0 relative error? It's not a good idea to use white for 0, as it can be confused with the paper color. Also, I find the colors to be too saturated, it's not easy to distinguish anything. Perhaps some lines "relative error = 0.8 - 0.5 - 0.2, etc" could be useful to separate?
- On Figures 2 a) and c), do we see confidence intervals? How are they computed? Would be nice to find this information on the paper.

Open questions/suggestions:
- I am curious if, in real world cases, learning a single distance for a population works well. My research is on learning individual preference, and the users' preferences are not necessarily represented with the same functions (as the preferences' criteria may be considered differently, sometimes with interaction, sometimes not, etc.).
- Does noise include possible incoherence preferences (the users incorrectly stating they prefer x to y related to their preferences)? Can your approach somehow detect possible incoherencies, or as shown in the last experiment, we will end up with a wrong approximation of M?
- Could you realistically do more than 30 runs? Wouldn't more runs reduce even more the variability of the results (even if, given what I suppose are the confidence intervals, the results are clear enough).
- Do you think it would be possible to have additional experiments to see the increase of the relative error given d? To see how much scalable to higher dimensions your approach is. The choice of m/number of users/number of subspaces can be optimized for each dimension I suppose.

Typos (I found some, even by not fully checking for them):
- In Page 3, you often said "Appendix C" instead of "Appendix B" for the algorithms.
- Algorithm 5, you forgot to add the annex link.
- Avoid writing "we've" (found it thrice).
- D1, second line "are have pairwise generic relations".

**Q9 Complying With Reviewing Instructions:**

Yes

---

> ### Author Rebuttal · Authors · 2024-04-07
>
> Thank you for your insightful and supportive feedback. We will make a pass over the paper to incorporate your detailed suggestions in the final version of our paper. We will open source our code and make sure to improve the readability of the figures.
>
> We follow up on some of your questions below:
>
> *1: I am curious if, in real world cases, learning a single distance for a population works well.*
>
> Information theoretically, since a Mahalanobis distance has $d(d+1)/2$ degrees of freedom, learning a personalized distance metric may not be feasible unless each user provides $\Omega{d^2}$ comparisons. In other words, in the data-starved regime, there is not much we can do in terms of individualized learning.
>
> This has been corroborated by an experiment in prior work [Canal et al., 2022]. On a dataset of color preferences, learning a common metric leads to competitive performance in comparison with learning individual metrics, in terms of the accuracy in predicted user responses. The performance of learning a common metric can be even better when each user answers less than 20 queries. See Figure 1d of [Canal et al., 2022] for details.
>
> &nbsp;
>
> *2: Does noise include possible incoherence preferences (the users incorrectly stating they prefer x to y related to their preferences)?*
>
> The noise model allows for misreported preferences. In fact, this simple model reflects human psychology [Coombs, 1964; Revelle, 2009]. When presented with two items to compare, our response is less noisy when a clear preference ranking exists. Conversely, when we are ambivalent between the two items, our response tends to be more random.
>
> - Coombs, C. H. (1964). A theory of data. Wiley.
> - Revelle, W. (2009). An introduction to psychometric theory with applications in R.
>
> &nbsp;
>
> *3: On Figures 2 a) and c), do we see confidence intervals? How are they computed? Would be nice to find this information on the paper.*
>
> The error bars in (a) and (c) represent one standard deviation from the mean. It’s in the caption now, and we will add this detail in the main paper for further clarity.
>
> &nbsp;
>
> *4: Do you think it would be possible to have additional experiments to see the increase of the relative error given d? To see how much scalable to higher dimensions your approach is.*
>
> For a fixed subspace dimension $r$, the recovery of subspace metrics is independent of the ambient dimension $d$, as shown in Remark 19. However, recovery errors of the full metric may increase depending on the number of subspaces. This was illustrated in Figure 2b and Figure G.1b of the paper: If we fix a row corresponding to a specific number of subspaces, then the relative errors increase as we move along the columns to the right, i.e., the ambient dimension $d$ increases.

---

### Official Review · Reviewer_NQR8 · 2024-03-23

**Q2-1 Originality-Novelty:** 2
**Q2-2 Correctness-Technical Quality:** 3
**Q2-5 Clarity Of Writing:** 3

**Q1 Summary And Contributions:**

The paper proposes a learning approach for pairwise preference comparisons. The main contributions are as follows: they provide an impossibility result, that is to say nothing can be learned if the items are in general position; and, they propose a divide-and-conquer approach and provide recovery guarantees when the items exhibit low-rank subspace-clusterable structure (coming from real-world data), with an empirical validation.

**Q2-3 Extent To Which Claims Are Supported By Evidence:**

2: Fair: the main claims are somewhat supported by evidence (but the experimental evaluation may be weak, or does not match entirely with the claims, important baselines may be missing, proofs contain important ideas but lack rigor, algorithmic details are only discussed superficially, references are imprecise, assumptions are not sufficiently motivated or explicated, etc.).

**Q2-4 Reproducibility:**

2: Fair: key resources (e.g. proofs, code, data) are unavailable but key details (e.g. proof sketches, experimental setup) are sufficiently well-described for an expert to confidently reproduce the main results.

**Q3 Main Strengths:**

The quality of the presentation is overall fairly good. The paper is relatively well organized and therefore it is quite easy to follow. The topic is interesting and fits UAI’s domains. The paper proposes both algorithmic and theoretical results.

**Q4 Main Weakness:**

The experimental section is quite weak in my opinion. The findings are validated by using synthetic data, but the approach have been motivated by the real-world data and no comparison with other approaches is provided. The theoretical results suffer of a lack of comments and explanations.

**Q5 Detailed Comments To The Authors:**

Too many important information are presented in the supplementary material and seem to be crucial for a good understanding of the paper (for instance, the state-of-the-art for having the literature related to the work, some key points of the proofs, …).
For Section 5, it would be great to discuss the choice of the dimension $r$ of the subspace $V$ (corresponding to the low-rank).

**Q9 Complying With Reviewing Instructions:**

Yes

---

> ### Author Rebuttal · Authors · 2024-04-07
>
> Thank you for your insightful feedback. To follow up on your comments:
>
> *1: The experimental section is quite weak in my opinion. The findings are validated by using synthetic data, but the approach have been motivated by the real-world data and no comparison with other approaches is provided.*
>
> Our main motivation in this paper is to undergo a theoretical study on whether an unknown Mahalanobis distance can be recovered when each user provides $o(d)$ preference comparisons. This is done by proving an impossibility result and also recovery guarantees when the set of items exhibit low-dimensional structure. The experiments serve mainly as an empirical validation of our theoretical results. We plan to further evaluate our algorithms on real-world data in our next piece of work.
>
> &nbsp;
>
> *2: The theoretical results suffer of a lack of comments and explanations. Too many important information are presented in the supplementary material and seem to be crucial for a good understanding of the paper...*
>
> Thank you for the suggestion. We will move important details to the main paper, and incorporate additional explanations and discussion of our theoretical results in the final version of the paper.
>
> &nbsp;
>
> *3: For Section 5, it would be great to discuss the choice of the dimension r of the subspace V (corresponding to the low-rank).*
>
> The dimension $r_\lambda$ of each subspace $V_\lambda$ is an intrinsic property of the set of items, $\mathcal{X}$ (it is not chosen by the learner). We will clarify this in the final version of our paper.

---

### Official Review · Reviewer_iShq · 2024-03-25

**Q2-1 Originality-Novelty:** 2
**Q2-2 Correctness-Technical Quality:** 3
**Q2-5 Clarity Of Writing:** 3

**Q1 Summary And Contributions:**

In this work, the authors study metric learning from pairwise preference comparisons, focusing on the ideal point model where items are embedded into $R^d$, and a user prefers one item over another if it is closer to the user's latent ideal point. The objective is to learn the underlying metric with minimal pairwise queries. Building upon prior research by Canal et al. [2022], which suggested a requirement of $\Theta(d)$ queries per user for learning the Mahanabolis distance metric, the authors address the practical infeasibility of asking $O(d)$ queries per user in high-dimensional data scenarios. They demonstrate the impossibility of learning a metric with fewer than $o(d)$ queries and propose a solution for datasets residing on a union of low-dimensional subspaces, showing that it is feasible to recover the metric using queries linear in the dimension of the subspace.

**Q2-3 Extent To Which Claims Are Supported By Evidence:**

3: Good: the main claims are supported by convincing evidence (in the form of adequate experimental evaluation, proofs, (pseudo-)code, references, assumptions).

**Q2-4 Reproducibility:**

3: Good: key resources (e.g. proofs, code, data) are available and key details (e.g. proofs, experimental setup) are sufficiently well-described for competent researchers to confidently reproduce the main results.

**Q3 Main Strengths:**

-- The authors provide an important lower bound result arguing that with $o(d)$ queries per user, metric learning of Mahanabolis distance is impossible.

-- The authors make a very interesting observation in Lemma 8 that even if the ideal point doesn't lie in the same subspace, we can identify an equivalent point on the subspace without affecting the metric learning problem. This greatly simplifies the setting and reduces it to learning appropriate subspaces.

-- The paper is more or less well-written with clear motivation and explanations.

**Q4 Main Weakness:**

-- It heavily relies on prior works on learning subspaces studied in Canal et al. [2022].

-- It is not entirely clear why the noise model was chosen. A brief discussion about the model in Section 5 and subsequent guarantees would add to the importance of the results in that section.

-- The authors should also describe the scenario when the subspace assumption doesn't hold. I.e., is the assumption testable? If not, what would be the consequences of violation?

**Q5 Detailed Comments To The Authors:**

-- I would suggest the authors briefly describe the choice of noise models in the introduction. There's been an abrupt jump from noiseless to noisy settings in the contributions section.

-- I would also suggest the authors write their theorem statements based on the queries per user (as this was the main motivation), rather than the error in approximation of the metric itself.

**Q9 Complying With Reviewing Instructions:**

Yes

---

> ### Author Rebuttal · Authors · 2024-04-07
>
> Thank you for your insightful feedback. To follow up on your comments:
>
> *1: It heavily relies on prior works on learning subspaces studied in Canal et al.*
>
> We have highlighted our contributions more explicitly in this [comment](https://openreview.net/forum?id=VFf9pwPYeX&noteId=jXtkQzql3J).
>
> &nbsp;
>
> *2: It is not entirely clear why the noise model was chosen. A brief discussion about the model in Section 5 and subsequent guarantees would add to the importance of the results in that section.*
>
> Thank you for the suggestion. We chose a noise model in which binary responses are generated according to a link function. This is among the simplest noise models, and it realistically reflects human psychology [Coombs, 1964; Revelle, 2009]. When presented with two items to compare, our response is less noisy when a clear preference ranking exists. Conversely, when we are ambivalent between the two items, our response tends to be more random.
>
> This is also a standard model [e.g., Agresti 2015] and has been studied in various machine learning problems, such as metric learning [Mason et al., 2017], dueling bandits [Yue and Joachims, 2009], and reinforcement learning from human feedback [Wang et al., 2024].
>
> In the final version of our paper, we will add a paragraph or two in Section 5 to further describe this probabilistic model and the subsequent guarantees.
>
> - Coombs, C. H. (1964). A theory of data. Wiley.
> - Revelle, W. (2009). An introduction to psychometric theory with applications in R.
> - Agresti, A. (2015). Foundations of linear and generalized linear models. John Wiley & Sons.
> - Mason, B., Jain, L., & Nowak, R. (2017). Learning low-dimensional metrics. Advances in neural information processing systems, 30.
> - Yue, Y., & Joachims, T. (2009, June). Interactively optimizing information retrieval systems as a dueling bandits problem. In Proceedings of the 26th Annual International Conference on Machine Learning (pp. 1201-1208).
> - Wang, Y., Liu, Q., & Jin, C. (2024). Is RLHF More Difficult than Standard RL? A Theoretical Perspective. Advances in Neural Information Processing Systems, 36.
>
> &nbsp;
>
> *3: The authors should also describe the scenario when the subspace assumption doesn't hold. I.e., is the assumption testable? If not, what would be the consequences of violation?*
>
> Information theoretically, it is easy to distinguish between a set of items in general position and a set of items that lie on a union of low-dimensional subspaces. Computationally, one can approximately recover such low-rank structures using dictionary learning.
>
> In Section 6 of the paper, we also considered a setting where the items lie approximately on a union of subspaces. Algorithmically, one can find the best low-rank approximation of the items and work from there. In Figure 2c, we showed empirically that our divide-and-conquer approach still works reasonably well when the subspace noise is low. In our next piece of work, we plan to further study this more general setting both theoretically and empirically.
>
> &nbsp;
>
> *4: I would also suggest the authors write their theorem statements based on the queries per user (as this was the main motivation), rather than the error in approximation of the metric itself.*
>
> That’s a good idea. Thanks for the suggestion.

---

### Official Review · Reviewer_iCQ5 · 2024-03-27

**Q2-1 Originality-Novelty:** 2
**Q2-2 Correctness-Technical Quality:** 2
**Q2-5 Clarity Of Writing:** 3

**Q1 Summary And Contributions:**

This paper examines metric learning from preference comparisons in the context of the ideal point model, where users prefer items closer to their latent ideal item within a space equipped with an unknown Mahalanobis distance. Recent studies indicate that recovering the metric and ideal items is possible with sufficient comparisons per user. However, with fewer comparisons (o(d)), it was unclear if metric learning was still feasible, as individual ideal items could not be learned. The findings reveal that with o(d) comparisons, information about the metric is only obtainable with additional conditions. The author presents two main results: one is the impossibility results and then a possibility under the assumption of latent space representation.

**Q2-3 Extent To Which Claims Are Supported By Evidence:**

2: Fair: the main claims are somewhat supported by evidence (but the experimental evaluation may be weak, or does not match entirely with the claims, important baselines may be missing, proofs contain important ideas but lack rigor, algorithmic details are only discussed superficially, references are imprecise, assumptions are not sufficiently motivated or explicated, etc.).

**Q2-4 Reproducibility:**

2: Fair: key resources (e.g. proofs, code, data) are unavailable but key details (e.g. proof sketches, experimental setup) are sufficiently well-described for an expert to confidently reproduce the main results.

**Q3 Main Strengths:**

- The paper is easy to follow.
- The problem of simultaneously learning metrics and preferences is interesting.
- The impossibility result is interesting.

**Q4 Main Weakness:**

- The intuition regarding the second contribution is not clear. Why would learning Mahalanobis distance help?
- What are the real-life scenarios for when it would hold?
- What is M in (1)
- Does the lower dimensional subspace structure inherently assume that the problem was not with respect to multiple users at the first place?
- How hard is solving the optimization problems in Algorithm 2 in practice?

- What is the justification for Assumption 17?

**Q5 Detailed Comments To The Authors:**

N/A

**Q9 Complying With Reviewing Instructions:**

Yes

---

> ### Author Rebuttal · Authors · 2024-04-07
>
> Thank you for your insightful feedback and questions. To follow up on them:
>
> *1: The intuition regarding the second contribution is not clear. Why would learning Mahalanobis distance help?*
>
> Learning a Mahalanobis distance is a fundamental question within metric learning [Kulis, 2013]. It can be interpreted as learning a linear layer on top of a given representation. By considering the Mahalanobis distance, we can gain insight into the fundamental problem of learning from limited preference comparisons. Our second contribution, as outlined in Section 1, essentially characterizes when and how a Mahalanobis distance can be recovered when each user provides only $o(d)$ preference comparisons. Other distances induced by a non-linear layer would be a natural generalization, though this is beyond the scope of this paper.
>
> - Kulis, B. (2013). Metric learning: A survey. Foundations and Trends® in Machine Learning, 5(4), 287-364.
>
> &nbsp;
>
> *2: What are the real-life scenarios for when it would hold?*
>
> It is common for real-world datasets to exhibit low-dimensional subspace structure. The subspaces may represent various categories or classes of items, such as different genres of movies or music. See more examples in this [comment](https://openreview.net/forum?id=VFf9pwPYeX&noteId=jXtkQzql3J).
>
> &nbsp;
>
> *3: What is M in (1)?*
>
> In Eq. (1), $M$ is the matrix representation of the unknown Mahalanobis distance $\rho$. $||x - u||_M^2 = (x - u)^\top M (x-u)$ defined in the equation above. We will provide additional clarity on this in the final version of the paper.
>
> &nbsp;
>
> *4: Does the lower dimensional subspace structure inherently assume that the problem was not with respect to multiple users at the first place?*
>
> The low-dimensional subspace structure pertains to the set of items, $\mathcal{X}$, and is not directly associated with the users that we may query. We show that we can exploit the such structure in the set of items by using answers by multiple users for pairs of items on low-dimensional subspaces to recover local restrictions of the metric and then recover the full metric $\rho$.
>
> &nbsp;
>
> *5: How hard is solving the optimization problems in Algorithm 2 in practice?*
>
> There are three optimization problems in Algorithm 2, all of which are convex and can be solved efficiently.
>
> In line 2, to approximately recover a subspace metric, it suffices to solve a semidefinite program. There exist a variety of efficient algorithms and solvers for large-scale cone problems. In particular, we used CVXPY with the splitting conic solver (SCS).
>
> In line 3, we use ordinary least squares to solve a linear regression problem.
>
> The optimization problem in line 4 is again convex: to find the projection of a symmetric matrix $A$ onto the positive semidefinite cone, it suffices to compute the eigendecomposition of $A$ and discard the terms associated with negative eigenvalues (e.g., Section 8.1.1 of [Boyd and Vandenberghe, 2004]).
>
> - Boyd, S. P., & Vandenberghe, L. (2004). Convex optimization. Cambridge university press.
>
> &nbsp;
>
> *6: What is the justification for Assumption 17?*
>
> In Assumption 17, we considered a noise model in which binary responses are generated according to a link function. This is among the simplest noise models, and it realistically reflects human psychology [Coombs, 1964; Revelle, 2009]. When presented with two items to compare, our response is less noisy when a clear preference ranking exists. Conversely, when we are ambivalent between the two items, our response tends to be more random.
>
> This is a standard model [e.g., Agresti 2015] and has been studied in various machine learning problems, such as metric learning [Mason et al., 2017], dueling bandits [Yue and Joachims, 2009], and reinforcement learning from human feedback [Wang et al., 2024].
>
> Furthermore, we note that our recovery guarantee in Theorem 15 is independent of this assumption.
>
> - Coombs, C. H. (1964). A theory of data. Wiley.
> - Revelle, W. (2009). An introduction to psychometric theory with applications in R.
> - Agresti, A. (2015). Foundations of linear and generalized linear models. John Wiley & Sons.
> - Mason, B., Jain, L., & Nowak, R. (2017). Learning low-dimensional metrics. Advances in neural information processing systems, 30.
> - Yue, Y., & Joachims, T. (2009, June). Interactively optimizing information retrieval systems as a dueling bandits problem. In Proceedings of the 26th Annual International Conference on Machine Learning (pp. 1201-1208).
> - Wang, Y., Liu, Q., & Jin, C. (2024). Is RLHF More Difficult than Standard RL? A Theoretical Perspective. Advances in Neural Information Processing Systems, 36.

---

### Official Review · Reviewer_Ezhx · 2024-03-29

**Q2-1 Originality-Novelty:** 2
**Q2-2 Correctness-Technical Quality:** 4
**Q2-5 Clarity Of Writing:** 4

**Q1 Summary And Contributions:**

The paper studies the problem of learning an unknown Mahalonobis distance (metric) from preference comparison data. This essentially boils down to learning a $d \times d$ dimensional positive definite matrix $M$. The motivation behind this problem is that while foundation models provide embeddings for a large class of items, the way users perceive similarities amongst these items does not merely correspond to the Euclidean distance between these objects. A better fit can be obtained by considering a more general Mahalanobis distance. Specifically, they assume the ideal point model, a models studied previously in the literature. This model assumes that users' response to preference comparisons is in accordance with the Mahalanobis distance. Thus such comparison data provides information about the unknown Mahalanobis distance.

This paper builds on two recent papers, [Xu and Davenport, 2020] and  [Canal et al., 2022]. Previous work shows that to recover the unknown metric, it suffices to have $O(d)$ preference comparisons per user. This paper argues that in practice, $d$ is often large; hence, it is worth investigating whether the unknown metric can be recovered from even fewer samples per user. This paper first shows a negative result, i.e., that $\Omega(d)$ samples per user are needed in general. They then go on to argue that under the assumption that the items display some specific low-dimensional structure (specifically, they lie on the union of low-dimensional subspaces), it is possible to firstly recover projections of the unknown metric onto these low dimensional spaces with a few samples, and the full metric can be recovered by 'stitching together' these projections. The authors provide a theoretical recovery guarantee and demonstrate the efficacy of their method through experiments on synthetic data.

**Q2-3 Extent To Which Claims Are Supported By Evidence:**

4: Excellent: all claims are supported by very convincing evidence (in the form of comprehensive experimental evaluation, rigorous mathematical proofs, detailed (pseudo-)code, precise references, well-motivated and realistic assumptions) and the authors deliver what they promise.

**Q2-4 Reproducibility:**

4: Excellent: key resources (e.g. proofs, code, data) are available and key details (e.g. proof sketches, experimental setup) are comprehensively described for competent researchers to confidently and easily reproduce the main results.

**Q3 Main Strengths:**

The main strengths are:

1. The paper studies an interesting research problem, which has been studied by recent papers.
2. The paper addresses an important problem left open by the literature.
3. The paper is well written. It was easy to understand what is the main problem the paper is tackling, and the main contributions of the paper. There are adequate references given.
4. The paper provides a clear model and a clear algorithm for solving the problem of interest.
5. The paper justifies the necessity for some additional assumptions by proving an impossibility result for the general case.
6. The paper's proposed algorithm is intuitive to understand and is presumably the best one can doo, given the problem formulation.

**Q4 Main Weakness:**

In a nutshell, the main weakness of the paper is its limited novelty. I elaborate as follows:

The paper is largely theoretical, with a few simulation results at the end. While the paper does provide a clear model of the data, a clear algorithm, and recovery guarantees for the algorithm, I feel the results of this paper are not significantly novel, given the past work in the area. In particular, I feel that this paper draws heavily from [Canal et al., 2022] on all three aspects: problem formulation, the algorithm, and the proofs.

Among the differences from [Canal et al., 2022], the main difference is introducing the notion of the union of subspaces over the item space. However, as this paper itself shows, this structure is not a new idea. Moreover, as they argue in Section 4, this is a particularly convenient assumption to make as it makes the task of learning the Mahalanobis distance quite easy: one can simply "stitch the metric over subspaces" to recover the full metric. The paper also does not justify whether this structure is indeed present in practice. The references given pertain to manifold learning, compressed sensing, and sparse coding, whereas this work is more pertinent to recommender systems. It is not at all clear whether the universe of items on Amazon or the collection of all songs on Spotify lie in a union of low-dimensional subspaces.

This paper could have made a strong case for itself if:
1. The extension beyond [Canal et al., 2022] led to some nice theoretical problems which require some novelty to solve. However, this is not the case. In fact, the contribution of this paper is to identify that the union of low-dimensional subspace case is a straightforward extension of the general case.
2. The paper had shown significant improvement in practical terms over the algorithm of [Canal et al., 2022]. The best way to do so would be to run both algorithms on real comparison data, for which there are many datasets. In fact, here, the paper does not even compare their algorithm with [Canal et al. 2022] on simulated data.
3. The paper argues that it is impractical to apply the algorithm of [Canal et al. 2022] on real data due to its $O(d)$ sample complexity per user, coupled with the fact that the embeddings produced by foundation models are high dimensional. However, a thorough critique of this point would have made the paper stronger. In particular:
a. can some dimensionality reduction technique not be used on the item features? How does that affect performance?
b. is it indeed more expensive to gain more samples from a single user than the same number of samples over different users? If the data is gathered from an MTurk like platform, the cost is proportional to the total human time spent, which should be similar across these two cases.
c. What if users demonstrated some special structure, such as clusters? Can't many users then be effectively treated as one?

**Q5 Detailed Comments To The Authors:**

Most of my comments are addressed in the points above. One additional point I would like to make is the following:

If there is genuine novelty (or complexity) in the design and analysis of the algorithms provided in this paper, they should be highlighted appropriately in the main part of the paper.

**Q9 Complying With Reviewing Instructions:**

Yes

---

> ### Author Rebuttal · Authors · 2024-04-07
>
> Thank you for your providing such constructive feedback. To follow up on some points:
>
> *1: This paper could have made a strong case for itself if the extension beyond [Canal et al., 2022] led to some nice theoretical problems which require some novelty to solve. However, this is not the case. In fact, the contribution of this paper is to identify that the union of low-dimensional subspace case is a straightforward extension of the general case.*
>
> We have highlighted our contributions more explicitly in this [comment](https://openreview.net/forum?id=VFf9pwPYeX&noteId=jXtkQzql3J), as suggested.
>
> One of the main contributions of our work is addressing the fundamental question of whether it is possible to learn a metric in $d$ dimensions with $o(d)$ queries per individual. This setting is not addressed by the results in [Canal et al., 2022]. We prove an impossibility result and provide simplified geometric characterizations for when learning a (subspace) metric is possible. Furthermore, we also provide a novel characterization of the identifiability of a (subspace) metric; in comparison with [Canal et al., 2022], our framing in terms of quadratic spanning (Definition 9) can be more succinct and intuitive.
>
> Additionally, it is a practical strength that our recovery framework (Algorithm 1) can exploit the union of low-dimensional subspace structure by making use of any metric learning algorithm, including but not limited to [Canal et al., 2022]. Furthermore, our focus on subspace structure is practically relevant since it is commonly exhibited in real-world datasets; we’ve included some additional examples in this [comment](https://openreview.net/forum?id=VFf9pwPYeX&noteId=jXtkQzql3J).
>
> &nbsp;
>
> *2: Is it indeed more expensive to gain more samples from a single user than the same number of samples over different users? If the data is gathered from an MTurk like platform, the cost is proportional to the total human time spent, which should be similar across these two cases.*
>
> Besides monetary expenses, there are a variety of reasons for which it is not possible to gather many responses from individual users: cognitive overload, privacy concerns, or lack of control over how many responses we can obtain from a user (e.g., the responses are inferred from the interaction of a user on a website). This is a problem—our impossibility result shows that generally, a user’s feedback provides no information until they have provided at least $d$ responses. It is therefore of practical interest to be able to make inferences even if we can collect far fewer responses per user than $d$.
>
> &nbsp;
>
> *3: This paper could have made a strong case for itself if the paper had shown significant improvement in practical terms over the algorithm of [Canal et al., 2022]. The best way to do so would be to run both algorithms on real comparison data, for which there are many datasets. In fact, here, the paper does not even compare their algorithm with [Canal et al. 2022] on simulated data.*
>
> Thank you for this suggestion. Here is a comparison [link](https://imgur.com/a/19SBHJt).  The experimental setup is the same as that for Figure 2a in the paper, with parameters set to $d = 5$, $n = 30$, and $r=1$. As the optimization problem in [Canal et al., 2022] is not designed to exploit the subspace-clusterability of the data, it performs much worse.
>
> &nbsp;
>
> *4: What if users demonstrated some special structure, such as clusters? Can't many users then be effectively treated as one?*
>
> It is definitely of interest to investigate other structural assumptions, e.g., user cluster structure, as you have suggested. While this is beyond the scope of this particular paper, we are excited to continue to develop this line of research in future works.

---

### Meta-Review · Area_Chair_B2ta · 2024-04-16

This paper studies the problem of learning a $d$-dimensional metric embedding from pairwise comparisons. It shows an impossibility result for learning the embedding with $o(d)$ queries per user, thereby settling an open question in the literature. It then reexamines this problem under a more restrictive assumption: that items lie in a union of subspaces within $R^d$. They show that under this assumption, $o(d)$ queries per user can be sufficient to learn the representations, and they provide evidence via a concrete algorithm and experiments using synthetic data.

The reviewers all recommend acceptance with various degrees of enthusiasm (from boderline accept to strong accept). There is general agreement that the paper is technically sound and well written overall, with some minor suggestions for improvements that the authors promised to heed (including a better comparison with [Canal et al 2022] at least on synthetic data). They also agree that the negative result is useful and important, while the second contribution caused slightly more controversy. The main questions concern the applicability of the assumption of subspace-cluster structure, and the absence of any evaluation on real data. Given that some public datasets are readily available, this would have dispelled any doubt that their model has practical relevance for the specific problem at hand.

The authors provided detailed and convincing responses to most reviewer concerns, with the exception of adding experiments over real data, which would have really strengthened the contribution. Nevertheless, I think that this paper is a very useful addition to the literature on comparison-based metric learning and should inspire interesting further research and debate.